# The ANTsX ecosystem for mapping the mouse brain

Nicholas J. Tustison [1] ✉, Min Chen[2], Fae N. Kronman [3], Jeffrey T. Duda [2], Clare Gamlin[4], Mia G. Tustison[5], Michael Kunst[4], Rachel Dalley [4], Staci Sorenson [4], Quanxin Wang[4], Lydia Ng[4], Yongsoo Kim [3] & James C. Gee [2] ✉

Large-scale efforts by the BRAIN Initiative Cell Census Network (BICCN) are generating a comprehensive reference atlas of cell types in the mouse brain. A key challenge in this effort is mapping diverse datasets, acquired with varied imaging, tissue processing, and profiling methods, into shared coordinate frameworks. Here, we present mouse brain mapping pipelines developed using the Advanced Normalization Tools Ecosystem (ANTsX) to align MERFISH spatial transcriptomics and high-resolution fMOST morphology data to the Allen Common Coordinate Framework (CCFv3), and developmental MRI and LSFM data to the Developmental CCF (DevCCF). Simultaneously, we introduce two novel methods: 1) a velocity field–based approach for continuous interpolation across developmental timepoints, and 2) a deep learning framework for automated brain parcellation using minimally annotated and publicly available data. All workflows are open-source and reproducible. We also provide general guidance for selecting appropriate strategies across modalities, enabling researchers to adapt these tools to new data.

Over the past decade, there have been significant advancements in mesoscopic single-cell analysis of the mouse brain. It is now possible to track single neurons[1], observe whole-brain developmental changes at cellular resolution[2], associate brain regions with genetic composition[3], and locally characterize neural connectivity[4]. These scientific achievements have been propelled by high-resolution profiling and imaging techniques that enable submicron, multimodal, 3D characterizations of whole mouse brains. Among these are micro-optical sectioning tomography[5,6], tissue clearing methods[1,7], spatial transcriptomics[8,9], and single-cell genomic profiling[10], each offering expanded specificity and resolution for cell-level brain analysis.

Recent efforts by the NIH BRAIN Initiative have mobilized large-scale international collaborations to create a comprehensive reference database of mouse brain structure and function. The BRAIN Initiative Cell Census Network has aggregated over 40 multimodal datasets from more than 30 research groups[11], many of which are registered to standardized anatomical coordinate systems to support integrated analysis. Among the most widely used of these frameworks is the Allen Mouse Brain Common Coordinate Framework (CCFv3)[12]. Other CCFs include modality-specific refs. [13–15] and developmental atlases[16,17] that track structural change across time.

Robust mapping of cell type data into CCFs is essential for integrative analysis of morphology, connectivity, and molecular identity. However, each modality poses unique challenges. For example, differences in tissue processing, imaging protocols, and anatomical completeness often introduce artifacts such as distortion, tearing, holes, and signal dropout[18–23]. Intensity differences and partial representations of anatomy can further complicate alignment. Also, while alternative strategies for mapping single-cell spatial transcriptomic data exist (e.g., gene expression–based models such as Tangram[24]) this work focuses on image-based anatomical alignment to common coordinate frameworks using spatially resolved reference images.

---

[1]Department of Radiology and Medical Imaging, University of Virginia, Charlottesville, VA, USA. [2]Department of Radiology, University of Pennsylvania, Philadelphia, PA, USA. [3]Department of Neuroscience and Experimental Therapeutics, Penn State University, Hershey, PA, USA. [4]Allen Institute for Brain Science, Seattle, WA, USA. [5]Santiago High School, Corona, CA, USA. ✉e-mail: ntustison@virginia.edu; gee@upenn.edu

Given this diversity specialized strategies are often needed to address the unique, modality-specific challenges.

Existing mapping solutions fall into three broad categories. The first includes integrated processing platforms that provide users with mapped datasets (e.g., Allen Brain Cell Atlas[25], Brain Architecture Portal[26], OpenBrainMap[27], and Image and Multi-Morphology Pipeline[28]). These offer convenience and high-quality curated data, but limited generalizability and customization. The second category involves highly specialized pipelines tailored to specific modalities such as histology[29–31], magnetic resonance imaging (MRI)[32–34], microCT[35,36], light sheet fluorescence microscopy (LSFM)[37,38], fluorescence micro-optical sectioning tomography (fMOST)[15,39], and spatial transcriptomics, including multiplexed error-robust fluorescence in situ hybridization (MERFISH)[40–42]. While effective, these solutions often require extensive engineering effort to adapt to new datasets or modalities. Finally, general-purpose toolkits such as elastix[43], Slicer3D[44], and the Advanced Normalization Tools Ecosystem (ANTsX)[45] have all been applied to mouse brain mapping scenarios. These toolkits support modular workflows that can be flexibly composed from reusable components, offering a powerful alternative to rigid, modality-specific solutions. However, their use often requires familiarity with pipeline modules, parameter tuning, and tool-specific conventions which can limit adoption.

Building on this third category, we describe a set of modular, ANTsX-based pipelines specifically tailored for mapping diverse mouse brain data into standardized anatomical frameworks. These include two new pipelines: a velocity field–based interpolation model that enables continuous transformations across developmental timepoints of the DevCCF, and a template-based deep learning pipeline for whole brain segmentation (i.e., brain extraction) and structural anatomical regional labeling of the brain (i.e., brain parcellation) requiring minimal annotated data. In addition, we include two modular pipelines for aligning MERFISH and fMOST datasets to the Allen CCFv3. While the MERFISH dataset was previously published as part of earlier BICCN efforts[46], the full image processing and registration workflow had not been described in detail until now. The fMOST workflow, by contrast, was developed internally to support high-resolution morphology mapping and has not been previously published in any form. Both pipelines were built using ANTsX tools, adapted for collaborative use with the Allen Institute, and are now released as fully reproducible, open-source workflows to support reuse and extension by the community. To facilitate broader adoption, we also provide general guidance for customizing these strategies across imaging modalities and data types. We first introduce key components of the ANTsX toolkit, which provide a basis for all of the mapping workflows described here, and then detail the specific contributions made in each pipeline.

The Advanced Normalization Tools Ecosystem (ANTsX) has been used in a number of applications for mapping mouse brain data as part of core processing steps in various workflows[31,46–49], particularly its pairwise, intensity-based image registration capabilities[50] and bias field correction[51]. Historically, ANTsX development is based on foundational approaches to image mapping[52–54], especially in the human brain, with key contributions such as the Symmetric Normalization (SyN) algorithm[50]. It has been independently evaluated in diverse imaging domains including multi-site brain MRI[55], pulmonary CT[56], and multi-modal brain tumor registration[57]. More recent contributions for mouse-specific applications showcase multimodal template generation[16] and anatomy-aware registration ANTsX functionality.

Beyond registration, ANTsX provides functionality for template generation[58], segmentation[59], preprocessing[51,60], and deep learning[45]. It has demonstrated strong performance in consensus labeling[61], brain tumor segmentation[62], and cardiac motion estimation[63]. Built on the Insight Toolkit (ITK)[64], ANTsX benefits from open-source contributions while supporting continued algorithm evaluation and innovation. In the context of mouse brain data, ANTsX provides a robust platform for developing modular pipelines to map diverse imaging modalities into CCFs. These tools span multiple classes of mapping problems: cross-modality image registration, landmark-driven alignment, temporal interpolation across developmental stages, and deep learning–based segmentation. As such, they also serve as illustrative case studies for adapting ANTsX tools to other use cases. We describe both shared infrastructure and targeted strategies adapted to the specific challenges of each modality. This paper highlights usage across distinct BICCN projects such as spatial transcriptomic data from MERFISH, structural data from fMOST, and multimodal developmental data from LSFM and MRI.

We introduce two novel contributions to ANTsX developed as part of collaborative efforts in creating the Developmental Common Coordinate Framework (DevCCF)[16]. First, we present an open-source velocity field–based interpolation framework for continuous mapping across the sampled embryonic and postnatal stages of the DevCCF atlas[16]. This functionality enables biologically plausible interpolation between timepoints via a time-parameterized diffeomorphic velocity model[65], inspired by previous work[66]. Second, we present a deep learning pipeline for structural parcellation of the mouse brain from multimodal MRI data. This includes two novel components: 1) a template-derived brain extraction model using augmented data from two ANTsX-derived template datasets[67,68], and 2) a template-derived parcellation model trained on DevCCF P56 labelings mapped from the AllenCCFv3. This pipeline demonstrates how ANTsX tools and public resources can be leveraged to build robust anatomical segmentation pipelines with minimal annotated data. We independently evaluate this framework using a longitudinal external dataset[69], demonstrating generalizability across specimens and imaging protocols. All components are openly available through the R and Python ANTsX packages, with general-purpose functionality documented in a reproducible, cross-platform tutorial (https://tinyurl.com/antsxtutorial). Code specific to this manuscript, including scripts to reproduce the novel contributions and all associated evaluations, is provided in a dedicated repository (https://github.com/ntustison/ANTsXMouseBrainMapping). Additional tools for mapping spatial transcriptomic (MERFISH) and structural (fMOST) data to the AllenCCFv3 are separately available at (https://github.com/dontminchenit/CCFAlignmentToolkit).

## Results

### Mapping multiplexed error-robust fluorescence in situ hybridization (MERFISH)

We developed an ANTsX-based pipeline to map spatial transcriptomic MERFISH data into the AllenCCFv3 (Fig. 1a). This approach was used in recent efforts to create a high-resolution transcriptomic atlas of the mouse brain[46]. The pipeline maps spatial gene expression patterns from MERFISH onto anatomical labels in the AllenCCFv3. It includes MERFISH-specific preprocessing steps such as section reconstruction, label generation from spatial transcriptomic maps, and anatomical correspondence mapping. Alignment proceeds in two stages: 1) 3D affine registration and section matching of the AllenCCFv3 to the MERFISH data, and 2) linear + deformable 2D section-wise alignment between matched MERFISH and atlas slices. These transformations are concatenated to produce a complete mapping from each MERFISH data to AllenCCFv3.

MERFISH imaging was performed on cryosectioned brains from C57BL/6 mice using previously described protocols[46]. Brains were placed into an optimal cutting temperature (OCT) compound (Sakura FineTek 4583) stored at −80°. The fresh frozen brain was sectioned at 10 $\mu m$ on Leica 3050 S cryostats at intervals of 200 $\mu m$ to evenly cover the brain. A set of 500 genes was selected to distinguish ~ 5200 transcriptomic clusters. Raw MERSCOPE data were decoded using Vizgen software (v231). Cell segmentation was performed using Cellpose[70,71] based on DAPI and PolyT stains which was propagated to

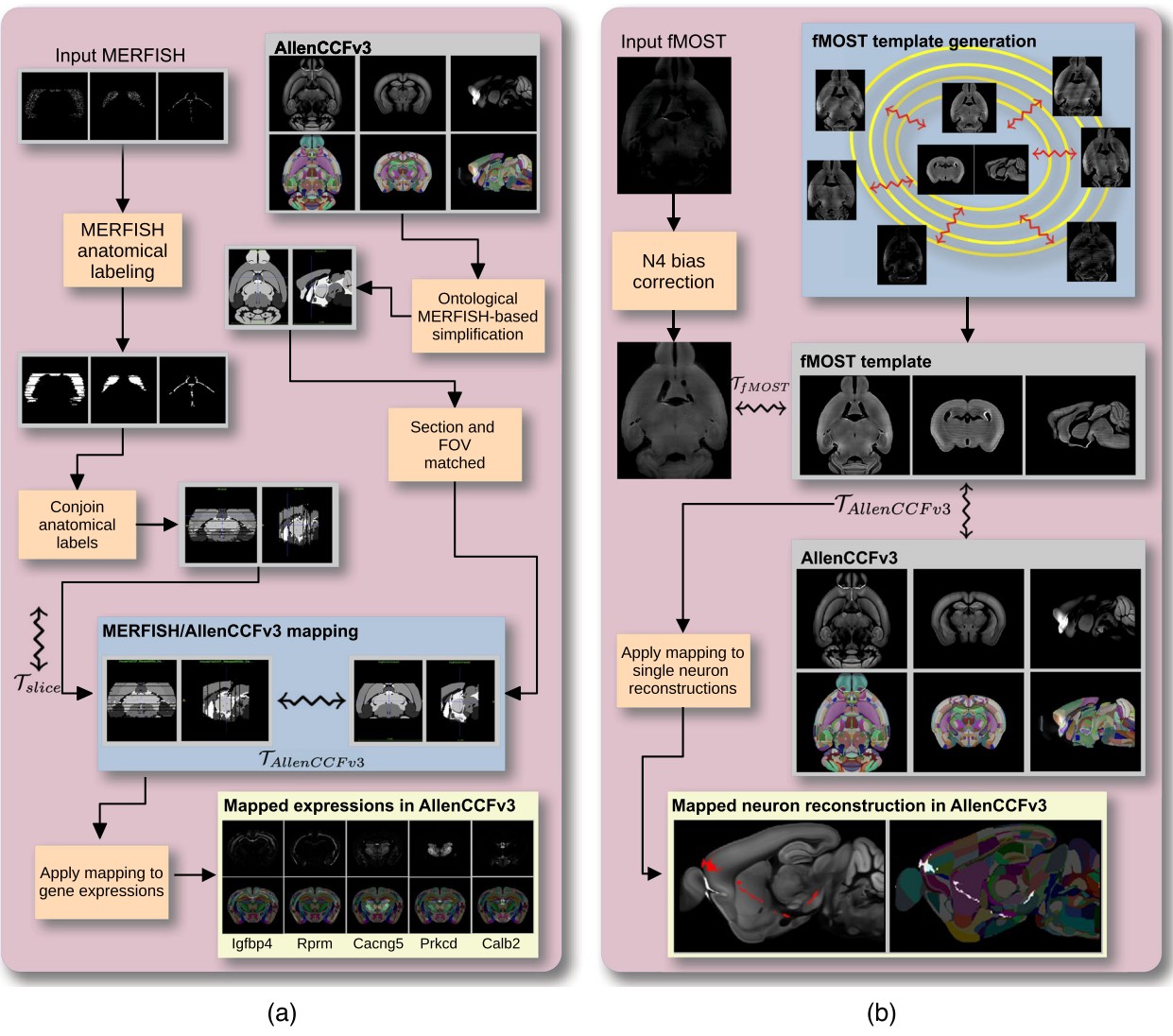

**Fig. 1 | Overview of ANTsX pipelines for mapping MERFISH and fMOST data to the AllenCCFv3.** Diagram of the two ANTsX-based pipelines for mapping (**a**) MERFISH and (**b**)fMOST data into the space of AllenCCFv3. Each generates the requisite transforms to map individual images to the CCF.

adjacent slices across z-planes. Each MERFISH cell was assigned a transcriptomic identity by mapping to a scRNA-seq reference taxonomy.

Alignment quality was evaluated iteratively by an expert anatomist, guided by expected gene-marker correspondences to AllenCCFv3 regions. As previously reported[46], further assessment of the alignment showed that, of the 554 terminal regions (gray matter only in the AllenCCFv3), only seven small subregions did not contain cells from the MERFISH dataset post registration: frontal pole, layer 1 (FRP1), FRP2/3, FRP5; accessory olfactory bulb, glomerular layer (AOBgl); accessory olfactory bulb, granular layer (AOBgr); accessory olfactory bulb, mitral layer (AOBmi); and accessory supraoptic group (ASO). A broader discussion of evaluation design choices and evaluation rationale is included in the Discussion.

## Mapping fluorescence micro-optical sectioning tomography (fMOST) data

We also constructed a pipeline for mapping fMOST images to the AllenCCFv3 using ANTsX (Fig. 1b). The approach leverages a modality-specific average fMOST atlas as an intermediate target, adapted from previous work in human and mouse brain mapping[12,15,16,58,72–75]. The atlas was constructed from 30 fMOST images selected to capture representative variability in anatomical shape and image intensity across the population. Preprocessing includes cubic B-spline downsampling to match the $25\,\mu m$ isotropic AllenCCFv3 resolution, stripe artifact suppression using a 3D notch filter implemented with SciPy's frequency-domain filtering tools, and N4 bias field correction[51]. A one-time, annotation-driven alignment registers the fMOST atlas to AllenCCFv3 using landmark-based registration of key structures. This canonical mapping is then reused. New fMOST specimens are first aligned to the fMOST atlas using standard intensity-based registration, and the concatenated transforms yield full spatial normalization to the AllenCCFv3. This same mapping can be applied to neuron reconstructions to facilitate population-level analysis of morphology and spatial distribution.

fMOST imaging was performed on 55 mouse brains with sparse transgenic labeling of neuron populations[76,77] using the high-throughput fMOST platform[78,79]. Voxel resolution was $0.35 \times 0.35 \times 1.0\,\mu m^3$. Two imaging channels were acquired: GFP-labeled neuron morphology (green), and propidium iodide counter-staining for cytoarchitecture (red). Alignment was performed using the red channel for its greater contrast, though multi-channel mapping is also supported.

The canonical mapping from the fMOST atlas to AllenCCFv3 was evaluated using both quantitative and qualitative approaches. Dice similarity coefficients were computed between corresponding

anatomical labels in the fMOST atlas and AllenCCFv3 following registration. These labels were manually annotated or adapted from existing atlas segmentations. Representative Dice scores included: whole brain (0.99), caudate putamen (0.97), fimbria (0.91), posterior choroid plexus (0.93), anterior choroid plexus (0.96), optic chiasm (0.77), and habenular commissure (0.63). In addition to these quantitative assessments, each registered fMOST specimen was evaluated qualitatively. An expert anatomist reviewed alignment accuracy and confirmed structural correspondence. Neuron reconstructions from individual brains were also transformed into AllenCCFv3 space, and their trajectories were visually inspected to confirm anatomical plausibility and preservation of known projection patterns. A broader discussion of evaluation design choices and evaluation rationale is included in the Discussion.

### Continuously mapping the DevCCF developmental trajectory

The DevCCF is an openly accessible resource for the mouse brain research community[16], comprising symmetric, multi-modal MRI and LSFM templates generated using the ANTsX framework[58]. It spans key stages of mouse brain development (E11.5, E13.5, E15.5, E18.5, P4, P14, and P56) and includes structural labels defined by a developmental ontology. The DevCCF was constructed in coordination with the AllenCCFv3 to facilitate integration across atlases and data types.

Although this collection provides broad developmental coverage, its discrete sampling limits the ability to model continuous transformations across time. To address this, we developed a velocity flow–based modeling approach that enables anatomically plausible, diffeomorphic transformations between any two continuous time points within the DevCCF range (Fig. 2). Unlike traditional pairwise interpolation, which requires sequential warping through each intermediate stage, this model, defined by a time-varying velocity field (i.e., a smooth vector field defined over space and time that governs the continuous deformation of an image domain), allows direct computation of deformations between any two time points in the continuum which improves smoothness and enables flexible spatiotemporal alignment. This functionality is implemented in both ANTsR and ANTsPy (see ants.fit_time_varying_transform_to_point_sets(...)) and integrates seamlessly with existing ANTsX workflows. The velocity field is represented as a 4D ITK image where each voxel stores the $x,y,z$ components of motion at a given time point. Integration of the time-varying velocity field uses uses $4^{th}$ order Runge-Kutta (ants.integrate_velocity_field(...))[80].

Each DevCCF template includes over 2500 labeled anatomical regions, with spatial resolutions ranging from 31.5 to $50\,\mu m$. For the velocity flow modeling task, we identified a common set of 26 bilateral regions (13 per hemisphere) that were consistently labeled across all timepoints. These regions span major developmental domains including the pallium, subpallium, midbrain, prosomeres, hypothalamus, hindbrain subregions, and key white matter tracts (Fig. 3).

Prior to velocity field optimization, all templates were rigidly aligned to the DevCCF P56 template using the centroids of these common label sets. Pairwise correspondence between adjacent timepoints was then computed using ANTsX's multi-metric registration via ants.registration(...). Instead of performing intensity-based multi-label registration directly, we constructed 24 binary label masks per atlas pair (one per structure) and optimized alignment using the mean squares similarity metric with the SyN transform[50].

To generate the point sets for velocity field optimization, we sampled both boundary (contour) and interior (region) points from the P56 labels and propagated them to each developmental stage using the learned pairwise transforms. Contours were sampled at 10% of available points and regions at 1%, yielding 173,303 total points per atlas ($N_{contour}$ = 98, 151; $N_{region}$ = 75,152). Boundary points were assigned double weight during optimization to emphasize anatomical boundary correspondence.

The velocity field was optimized using the seven corresponding point sets and their associated weights. The field geometry was defined at [256, 182, 360] with 11 integration points at $50\,\mu m$ resolution, yielding a compressed velocity model of ~ 2 GB. This resolution balanced accuracy and computational tractability while remaining portable. All data and code are publicly available in the accompanying GitHub repository.

To normalize temporal spacing, we assigned scalar values in [0, 1] to each template. Given the nonlinear spacing in postnatal development, we applied a logarithmic transform to the raw time values prior to normalization. Within this logarithmic temporal transform, P56 was assigned a span of 28 postnatal days to reflect known developmental dynamics (i.e., in terms of modeling the continuous deformation, the morphological changes between Day 28 and Day 56 are insignificant). This improved the temporal distribution of integration points (Fig. 4, right panel).

Optimization was run for a maximum of 200 iterations using a 2020 iMac (3.6 GHz 10-Core Intel Core i9, 64 GB RAM), with each iteration taking ~ 6 min. During each iteration, the velocity field was updated across all 11 integration points by computing regularized displacement fields between warped point sets at adjacent time slices. Updates were applied using a step size of $\delta = 0.2$. Convergence was assessed via average displacement error across all points, with final convergence achieved after ~ 125 iterations (Fig. 4, left panel). Median errors across integration points also trended toward zero, albeit at varying rates. To benchmark performance, we compared the velocity model's region-based alignment to traditional pairwise registration using SyN, a widely used diffeomorphic algorithm. The velocity model achieved comparable Dice scores at sampled timepoints while additionally offering smooth interpolation across the entire developmental trajectory.

Once optimized, the velocity field enables the computation of diffeomorphic transformations between any pair of continuous time points within the DevCCF developmental range. Figure 5 illustrates cross-warping between all DevCCF stages using the velocity flow

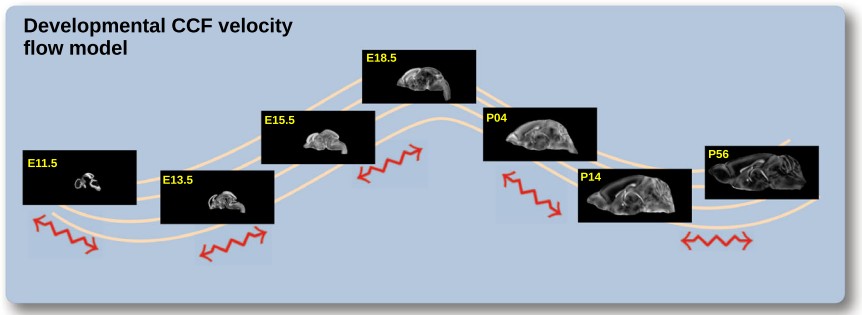

**Fig. 2 | Continuous developmental mapping enabled by the DevCCF velocity flow model.** The spatial transformation between any two time points within the continuous DevCCF longitudinal developmental trajectory is available through the use of ANTsX functionality for generating a velocity flow model.

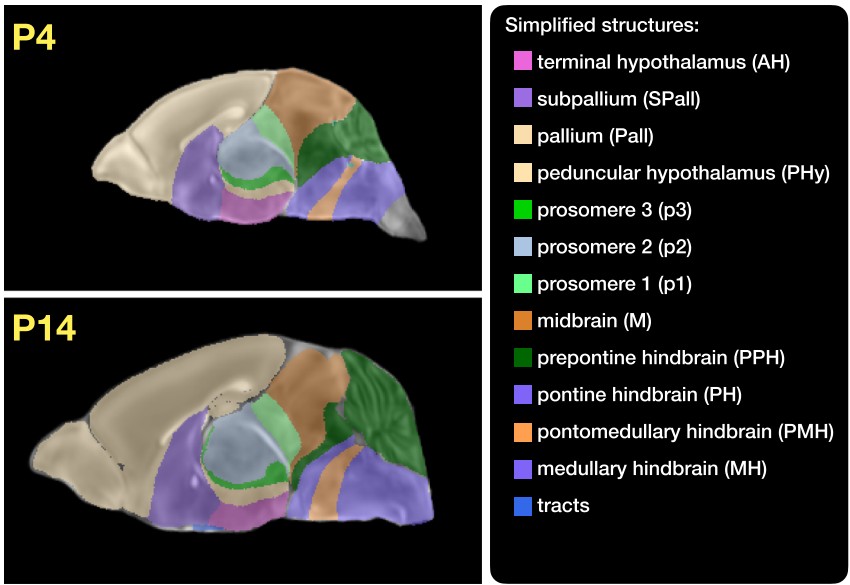

**Fig. 3 | Common anatomical labels across developmental stages of the DevCCF.** Annotated regions representing common labels across developmental stages, shown for both P4 and P14.

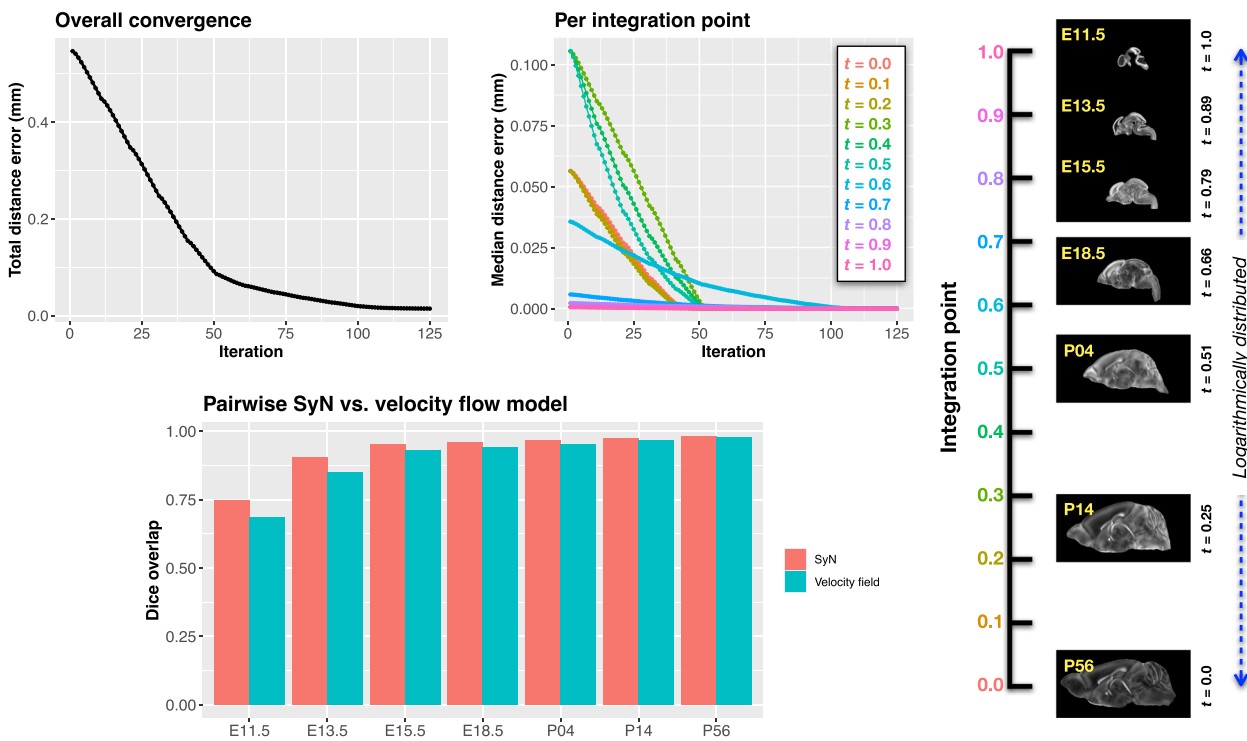

**Fig. 4 | Convergence and evaluation of the velocity flow model across the DevCCF developmental trajectory.** (Top left) Total displacement error over iterations. (Top right) Median displacement error per integration point across the optimization timeline, spanning embryonic (E11.5) to postnatal (P56) stages. (Bottom) Dice similarity scores comparing region-level label overlap between: (1) conventional pairwise SyN registration and (2) velocity flow-based deformation, across intermediate timepoints. Using region-based pairwise registration with SyN as a performance upper bound, the velocity flow model achieves comparable accuracy while also enabling smooth, continuous deformation across the full developmental continuum.

model. In addition to facilitating flexible alignment between existing templates, the model also supports the synthesis of virtual templates at intermediate, unsampled developmental stages. As shown in Fig. 6, we demonstrate the creation of virtual age templates (e.g., P10.3 and P20) by warping adjacent developmental atlases to a target timepoint and constructing an averaged representation using ANTsX's template-building functionality.

All usage examples, scripts, and supporting data for full reproducibility are publicly available in the associated codebase.

## Automated structural labeling of the mouse brain
Structural labeling strategies for the mouse brain are essential for understanding the organization and function of the murine nervous system[81]. By dividing the brain into anatomically or functionally

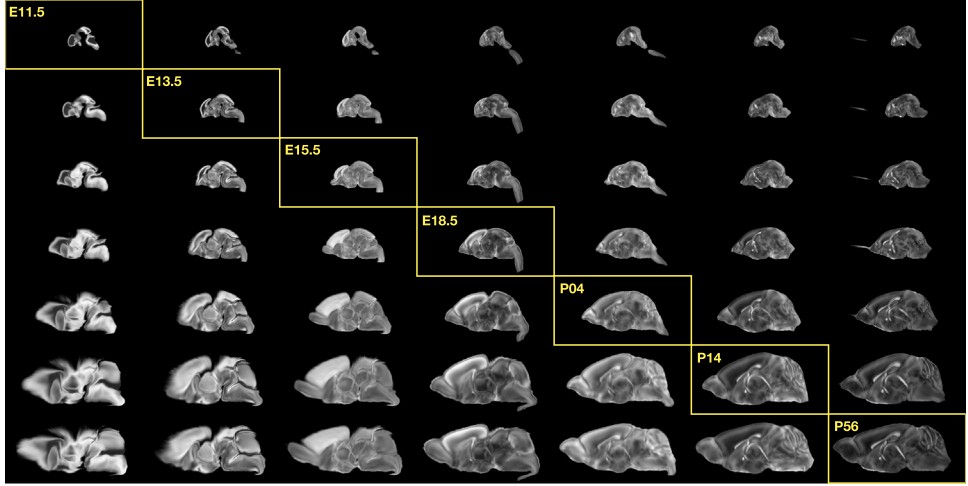

**Fig. 5 | Visualization of DevCCF templates warped across developmental time points.** Mid-sagittal visualization of DevCCF templates warped to every other time point. Each row is a reference space; each column is a warped input. Diagonal entries show original templates.

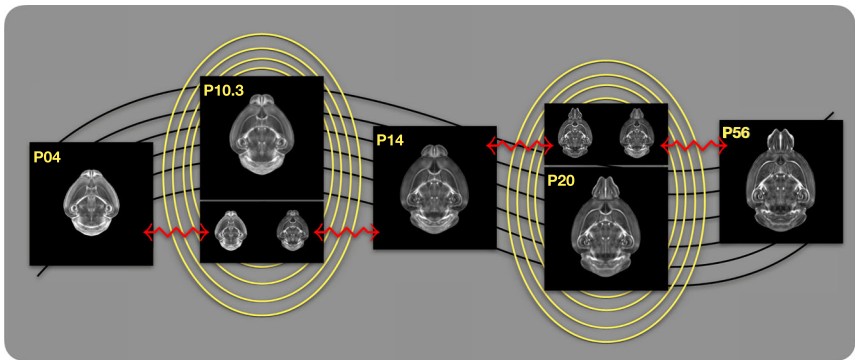

**Fig. 6 | Generation of virtual DevCCF templates at intermediate developmental stages.** Example of generating "virtual" DevCCF templates at intermediate time points (e.g., P10.3, P20) by warping adjacent stages to a shared time and averaging using ANTsX.

defined regions, researchers can localize biological processes, relate regional features to behavior, or quantify spatial variation in gene expression patterns[82,83]. While deep learning techniques have yielded robust segmentation and labeling tools for the human brain (e.g., SynthSeg[84], ANTsXNet[45]), analogous development for mouse data (e.g., MEMOS[85]) has been limited. Mouse neuroimaging often presents unique challenges, such as highly anisotropic sampling, that complicate transfer of existing tools. At the same time, high resolution resources like the AllenCCFv3 and DevCCF provide reference label sets that can serve as training data. We demonstrate how ANTsX can be used to construct a full structural labeling pipeline for the mouse brain (Fig. 7), including both whole brain segmentation (i.e., brain extraction) and the subsequent template-based region segmentation.

To develop a general-purpose mouse brain extraction model, we constructed whole-head templates from two publicly available T2-weighted datasets. The first dataset, from the Center for Animal MRI (CAMRI) at the University of North Carolina at Chapel Hill[67], includes 16 isotropic MRI volumes acquired at $0.16 \times 0.16 \times 0.16$ mm$^3$ resolution. The second dataset[68] comprises 88 specimens acquired in three orthogonal 2D views (coronal, axial, sagittal) at $0.08 \times 0.08$ mm$^2$ in-plane resolution with 0.5 mm slice thickness. These orthogonal 2D acquisitions were reconstructed into high-resolution 3D volumes using a B-spline fitting algorithm[86]. Using this synthesized dataset and the CAMRI images, we created two ANTsX-based population templates[58], each paired with a manually delineated brain mask. These served as the basis for training an initial template-based brain extraction model. Deep learning training of the network employed aggressive data

augmentation strategies, including bias field simulation, histogram warping, random spatial deformation, noise injection, and anisotropic resampling. This enabled the model to generalize beyond the two templates. The initial model was released through ANTsXNet and made publicly available.

Subsequent community use led to further improvements. A research group applying the tool to their own ex vivo T2-weighted mouse brain data contributed a third template and associated mask (acquired at 0.08 mm isotropic resolution). Incorporating this into the training data improved robustness and accuracy to an independent dataset and extended the model's generalizability. The refined model is distributed through ANTsPyNet via `antspynet.mouse_brain_extraction(...)`.

The AllenCCFv3 atlas and its hierarchical ontology, along with the DevCCF, provide a strong foundation for developing region-wise anatomical labeling models for multi-modal mouse brain imaging. Using the `allensdk` Python library, we generated a coarse segmentation scheme by grouping anatomical labels into six major regions: cerebral cortex, cerebral nuclei, brainstem, cerebellum, main olfactory bulb, and hippocampal formation. These labels were mapped onto the P56 T2-weighted DevCCF template to serve as training targets. We trained a 3D U-net–based segmentation network using this template and the same augmentation strategies described for brain extraction. The model is publicly available via ANTsXNet (`antspynet.mouse_brain_parcellation(...)`) and supports robust anatomical labeling across diverse imaging geometries and contrasts. The inclusion of aggressive augmentation, including simulated anisotropy,

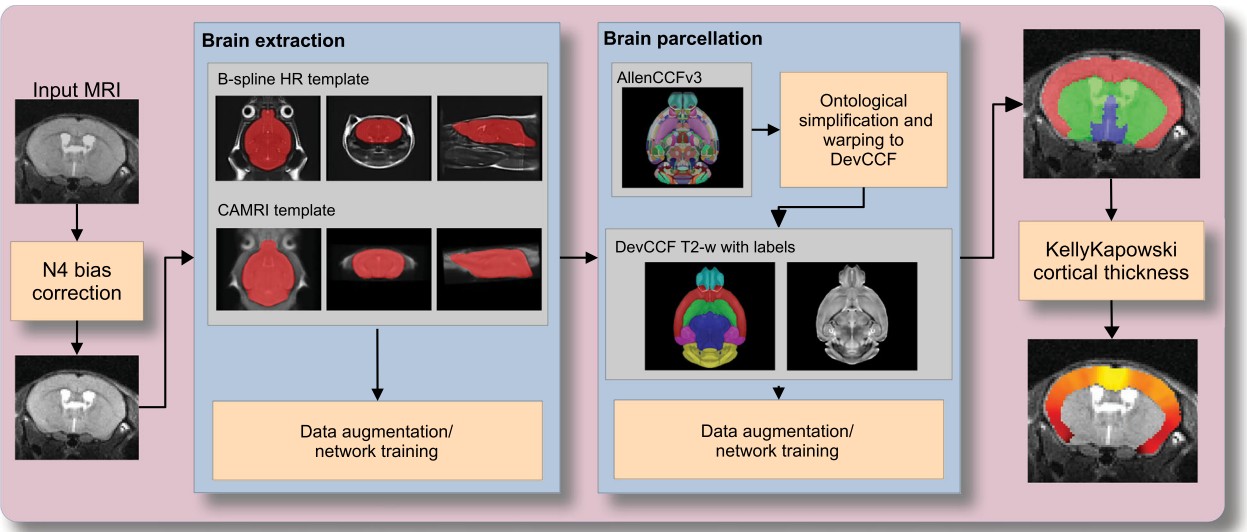

**Fig. 7 | Deep learning pipelines for mouse brain extraction and parcellation.** The mouse brain cortical labeling pipeline integrates two deep learning components for brain extraction and anatomical region segmentation. Both networks rely heavily on data augmentation applied to templates constructed from open datasets. The framework also supports further refinement or alternative label sets tailored to specific research needs. Possible applications include voxelwise cortical thickness estimation.

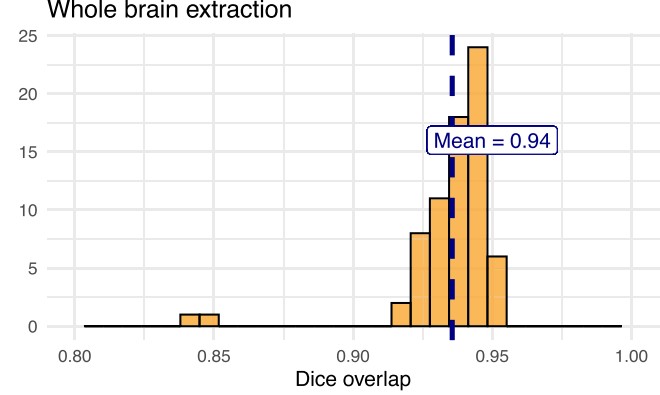

**Fig. 8 | Evaluation of ANTsX brain extraction across an independent dataset.** Evaluation of the ANTsX mouse brain extraction on an independent, publicly available dataset consisting of 12 specimens × 7 time points = 84 total images. Dice overlap comparisons with the user-generated brain masks provide good agreement with the automated results from the brain extraction network.

enables the model to perform well even on thick-slice input data. Internally, the model reconstructs isotropic probability and label maps, facilitating downstream morphometric analyses. For example, this network integrates with the ANTsX cortical thickness estimation pipeline (`antspynet.mouse_cortical_thickness(...)`) to produce voxelwise cortical thickness maps, even when applied to anisotropic or limited-resolution mouse brain data.

For evaluation, we used an additional publicly available dataset[69] that is completely independent from the data used in training the brain extraction and parcellation networks. Data includes 12 specimens each imaged at seven time points (Day 0, Day 3, Week 1, Week 4, Week 8, Week 20) with in-house-generated brain masks (i.e., produced by the data providers) for a total of 84 images. Spacing is anisotropic with an in-plane resolution of 0.1 × 0.1 mm² and a slice thickness of 0.5 mm.

Figure 8 summarizes the whole-brain overlap between manually segmented reference masks and the predicted segmentations for all 84 images in the evaluation cohort. The proposed network demonstrates excellent performance in brain extraction across a wide age range. To further assess the utility of the parcellation network, we used the predicted labels to guide anatomically informed registration to the AllenCCFv3 atlas using ANTsX multi-component registration, and compared this to intensity-only registration (Fig. 9). While intensity-based alignment performs reasonably well, incorporating the predicted parcellation significantly improves regional correspondence. Dice scores shown in Fig. 9c were computed using manually segmented labels transformed to AllenCCFv3 space.

## Discussion

The diverse mouse brain cell type profiles gathered through BICCN and associated efforts provide a rich multi-modal resource to the research community. However, despite significant progress, optimal leveraging of these valuable resources remains an ongoing challenge. A central component to data integration is accurately mapping novel cell type data into common coordinate frameworks (CCFs) for subsequent processing and analysis. To meet these needs, tools for mapping mouse brain data must be both broadly accessible and capable of addressing challenges unique to each modality. In this work, we described modular ANTsX-based pipelines developed to support three distinct BICCN efforts encompassing spatial transcriptomic, morphological, and developmental data. We demonstrated how a flexible image analysis toolkit like ANTsX can be tailored to address specific modality-driven constraints by leveraging reusable, validated components.

As part of collaborative efforts with the Allen Institute for Brain Science and the broader BICCN initiative, we developed two modular pipelines for mapping MERFISH and fMOST datasets to the AllenCCFv3. These workflows were designed to accommodate the specific requirements of high-resolution transcriptomic and morphological data while leveraging reusable components from the ANTsX ecosystem. The MERFISH pipeline incorporates preprocessing and registration steps tailored to known anatomical and imaging artifacts in multiplexed spatial transcriptomic data. While the general mapping strategy is applicable to other sectioned histological datasets, these refinements demonstrate how general-purpose tools can be customized to meet the demands of specialized modalities. The fMOST workflow, in contrast, emphasizes reusability and consistency across

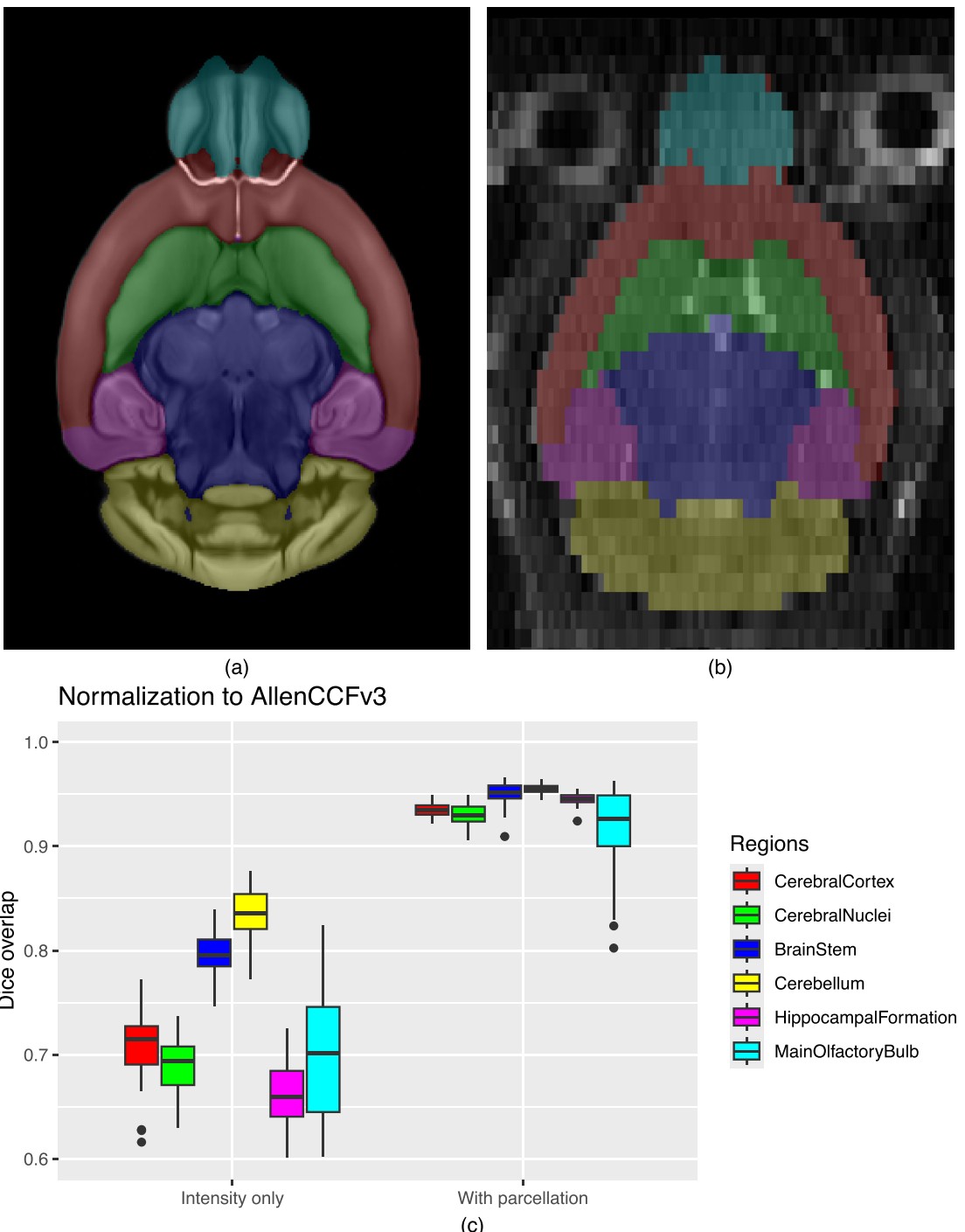

**Fig. 9 | Performance of ANTsX deep learning–based mouse brain parcellation.** Evaluation of the ANTsX deep learning--based mouse brain parcellation on a diverse MRI cohort. **a** T2-weighted DevCCF P56 template with the six-region parcellation: cerebral cortex, nuclei, brain stem, cerebellum, main olfactory bulb, and hippocampal formation. **b** Example segmentation result from a representative subject (NR5, Day 0) using the proposed deep learning pipeline. **c** Box plots show Dice overlap across subjects for each registration approach and region. The centre line is the median; box bounds are the interquartile range (25th--75th percentiles); whiskers extend to the minimum and maximum values within $1.5 \times$ IQR of the lower/upper quartiles; points beyond the whiskers are outliers.

large datasets. It introduces an intermediate, canonical fMOST atlas to stabilize transformations to the AllenCCFv3, reducing the need for repeated manual alignment and enabling standardized mapping of single-neuron reconstructions to a common coordinate framework.

Evaluation of both workflows followed established QA/QC protocols used at the Allen Institute, emphasizing biologically meaningful criteria such as expected gene-marker alignment (MERFISH) and accurate reconstruction of neuronal morphology (fMOST). These domain-informed assessments, also used in prior large-scale mapping projects[46], prioritize task-relevant accuracy over other possible benchmarks such as Dice coefficients or landmark distances. While formal quantitative scores were not reported for these specific pipelines, they both demonstrate reliable, expert-validated performance in collaborative contexts. Additional documentation and evaluation commentary are available in the updated CCFAlignmentToolkit GitHub repository.

For developmental data, we introduced a velocity field–based model for continuous interpolation between discrete DevCCF timepoints. Although the DevCCF substantially expands coverage of developmental stages relative to prior atlases, temporal gaps remain. The velocity model enables spatio-temporal transformations within the full developmental interval and supports the generation of virtual templates at unsampled ages. This functionality is built using ANTsX components for velocity field optimization and integration, and offers a novel mechanism for interpolating across the non-linear developmental trajectory of the mouse brain. Such interpolation has potential utility for both anatomical harmonization and longitudinal analyses. Interestingly, long-range transformations (e.g., P56 to E11.5) revealed anatomy evolving in plausible ways yet sometimes diverging from known developmental patterns (e.g., hippocampal shape changes) reflecting the input data and offering insight into temporal gaps. These behaviors could assist future efforts to determine which additional time points would most improve spatiotemporal coverage.

We also introduced a template-based deep learning pipeline for mouse brain extraction and parcellation using aggressive data augmentation. This approach is designed to reduce the reliance on large annotated training datasets, which remain limited in the mouse imaging domain. Evaluation on independent data demonstrates promising generalization, though further refinement will be necessary. As with our human-based ANTsX pipelines, failure cases can be manually corrected and recycled into future training cycles. Community contributions are welcomed and encouraged, providing a pathway for continuous improvement and adaptation to new datasets.

The ANTsX ecosystem offers a powerful foundation for constructing scalable, reproducible pipelines for mouse brain data analysis. Its modular design and multi-platform support enable researchers to develop customized workflows without extensive new software development. The widespread use of ANTsX components across the neuroimaging community attests to its utility and reliability. As a continuation of the BICCN program, ANTsX is well positioned to support the goals of the BRAIN Initiative Cell Atlas Network (BICAN) and future efforts to extend these mapping strategies to the human brain.

## Methods

The following methods are all available as part of the ANTsX ecosystem with analogous elements existing in both ANTsR (ANTs in R) and ANTsPy (ANTs in Python), underpinned by a shared ANTs/ITK C++ core. Most development for the work described was performed using ANTsPy. For equivalent functionality in ANTsR, we refer the reader to the comprehensive ANTsX tutorial: https://tinyurl.com/antsxtutorial.

### General ANTsX utilities

Although focused on distinct data types, the three pipelines presented in this work share common components that address general challenges in mapping mouse brain data. These include correcting image intensity artifacts, denoising, spatial registration, template generation, and visualization. Table 1 provides a concise summary of the relevant ANTsX functionality.

Standard preprocessing steps in mouse brain imaging include correcting for spatial intensity inhomogeneities and reducing image noise, both of which can impact registration accuracy and downstream analysis. ANTsX provides implementations of widely used methods for these tasks. The N4 bias field correction algorithm[51], originally developed in ANTs and contributed to ITK, mitigates artifactual, low-frequency intensity variation and is accessible via `ants.n4_bias_field_correction(...)`. Patch-based denoising[60] has been implemented as `ants.denoise_image(...)`.

ANTsX includes a robust and flexible framework for pairwise and groupwise image registration[80]. At its core is the SyN algorithm[50], a symmetric diffeomorphic model with optional B-spline

regularization[66]. In ANTsPy, registration is performed via `ants.registration(...)` using preconfigured parameter sets (e.g., `antsRegistrationSyNQuick[s]`, `antsRegistrationSyN[s]`) suitable for different imaging modalities and levels of computational demand. Resulting transformations can be applied to new images with `ants.apply_transforms(...)`.

ANTsX supports population-based template generation through iterative pairwise registration to an evolving estimate of the mean shape and intensity reference space across subjects[58]. This functionality was used in generating the DevCCF templates[16]. The procedure, implemented as `ants.build_template(...)`, produces average images in both shape and intensity by aligning all inputs to a common evolving template.

To support visual inspection and quality control, ANTsPy provides flexible image visualization with `ants.plot(...)`. This function enables multi-slice and multi-orientation rendering with optional overlays and label maps.

### Mapping fMOST data to AllenCCFv3

Mapping fMOST data into the AllenCCFv3 presents unique challenges due to its native ultra-high resolution and imaging artifacts common to the fMOST modality. Each fMOST image can exceed a terabyte in size, with spatial resolutions far exceeding those of the AllenCCFv3 (25 $\mu m$ isotropic). To reduce computational burden and prevent resolution mismatch, each fMOST image is downsampled using cubic B-spline interpolation via `ants.resample_image(...)` to match the template resolution.

Stripe artifacts (i.e., periodic intensity distortions caused by nonuniform sectioning or illumination) are common in fMOST and can mislead deformable registration algorithms. These were removed using a custom 3D notch filter (`remove_stripe_artifact(...)`) implemented in the `CCFAlignmentToolkit` using SciPy frequency domain filtering. The filter targets dominant stripe frequencies along a user-specified axis in the Fourier domain. In addition, intensity inhomogeneity across sections, often arising from variable staining or illumination, was corrected using N4 bias field correction.

To facilitate reproducible mapping, we first constructed a contralaterally symmetric average template from 30 fMOST brains and their mirrored counterparts using ANTsX template-building tools. Because the AllenCCFv3 and fMOST data differ substantially in both intensity contrast and morphology, direct deformable registration between individual fMOST brains and the AllenCCFv3 was insufficiently robust. Instead, we performed a one-time expert-guided label-driven registration between the average fMOST template and AllenCCFv3. This involved sequential alignment of seven manually selected anatomical regions: 1) brain mask/ventricles, 2) caudate/putamen, 3) fimbria, 4) posterior choroid plexus, 5) optic chiasm, 6) anterior choroid plexus, and 7) habenular commissure which were prioritized to enable coarse-to-fine correction of shape differences. Once established, this fMOST-template-to-AllenCCFv3 transform was reused for all subsequent specimens. Each new fMOST brain was then registered to the average fMOST template using intensity-based registration, followed by concatenation of transforms to produce the final mapping into AllenCCFv3 space.

A key advantage of fMOST imaging is its ability to support single neuron projection reconstruction across the entire brain[77]. Because these reconstructions are stored as 3D point sets aligned to the original fMOST volume, we applied the same composite transform used for image alignment to the point data using ANTsX functionality. This enables seamless integration of cellular morphology data into AllenCCFv3 space, facilitating comparative analyses across specimens.

### Mapping MERFISH data to AllenCCFv3

MERFISH data are acquired as a series of 2D tissue sections, each comprising spatially localized gene expression measurements at

**Table 1 | Sampling of ANTsX functionality**

| **ANTsPy: Preprocessing** | |
|---|---|
| bias field correction | `n4_bias_field_correction(...)` |
| image denoising | `denoise_image(...)` |
| ANTsPy: Registration | |
| intensity image registration | `registration(...)` |
| label image registration | `label_image_registration(...)` |
| image transformation | `apply_transforms(...)` |
| template generation | `build_template(...)` |
| landmark registration | `fit_transform_to_paired_points(...)` |
| time-varying landmark reg. | `fit_time_varying_transform_to_point_sets(...)` |
| integrate velocity field | `integrate_velocity_field(...)` |
| invert displacement field | `invert_displacement_field(...)` |
| ANTsPy: Segmentation | |
| MRF-based segmentation | `atropos(...)` |
| Joint label fusion | `joint_label_fusion(...)` |
| diffeomorphic thickness | `kelly_kapowski(...)` |
| ANTsPy: Miscellaneous | |
| Regional intensity statistics | `label_stats(...)` |
| Regional shape measures | `label_geometry_measures(...)` |
| B-spline approximation | `fit_bspline_object_to_scattered_data(...)` |
| Visualize images and overlays | `plot(...)` |
| ANTsPyNet: Mouse-specific | |
| brain extraction | `mouse_brain_extraction(...modality="t2"...)` |
| brain parcellation | `mouse_brain_parcellation(...)` |
| cortical thickness | `mouse_cortical_thickness(...)` |
| super resolution | `mouse_histology_super_resolution(...)` |

ANTsX provides state-of-the-art functionality for processing biomedical image data. Such tools, including deep learning networks, support a variety of mapping-related tasks. A more comprehensive listing of ANTsX tools with self-contained R and Python examples is provided as a gist page on GitHub (https://tinyurl.com/antsxtutorial).

subcellular resolution. To enable 3D mapping to the AllenCCFv3, we first constructed anatomical reference images by aggregating the number of detected transcripts per voxel across all probes within each section. These 2D projections were resampled to a resolution of 10 $\mu m$ × 10 $\mu m$ to match the in-plane resolution of the AllenCCFv3.

Sections were coarsely aligned using manually annotated dorsal and ventral midline points, allowing initial volumetric reconstruction. However, anatomical fidelity remained limited by variation in section orientation, spacing, and tissue loss. To further constrain alignment and enable deformable registration, we derived region-level anatomical labels directly from the gene expression data.

To assign region labels to the MERFISH data, we use a cell type clustering approach previously detailed[46]. In short, manually dissected scRNAseq data was used to establish the distribution of cell types present in each of the following major regions: cerebellum, CTXsp, hindbrain, HPF, hypothalamus, isocortex, LSX, midbrain, OLF, PAL, sAMY, STRd, STRv, thalamus and hindbrain. Clusters in the scRNA-seq dataset were then used to assign similar clusters of cell types in the MERFISH data to the regions they are predominantly found in the scRNA-seq data. To account for clusters that were found at low frequency in regions outside its main region we calculated for each cell its 50 nearest neighbors in physical space and reassigned each cell to the region annotation dominating its neighborhood.

A major challenge was compensating for oblique cutting angles and non-uniform section thickness, which distort the anatomical shape and spacing of the reconstructed volume. Rather than directly warping the MERFISH data into atlas space, we globally aligned the AllenCCFv3 to the MERFISH coordinate system. This was done via an affine transformation followed by resampling of AllenCCFv3 sections to match the number and orientation of MERFISH sections. This approach

minimizes interpolation artifacts in the MERFISH data and facilitates one-to-one section matching.

We used a 2.5D approach for fine alignment of individual sections. In each MERFISH slice, deformable registration was driven by sequential alignment of anatomical landmarks between the label maps derived from MERFISH and AllenCCFv3. A total of nine regions, including isocortical layers 2/3, 5, and 6, the striatum, hippocampus, thalamus, and medial/lateral habenula, were registered in an empirically determined order. After each round, anatomical alignment was visually assessed by an expert, and the next structure was selected to maximize improvement in the remaining misaligned regions.

The final transform for each section combined the global affine alignment and the per-structure deformable registrations. These were concatenated to generate a 3D mapping from the original MERFISH space to the AllenCCFv3 coordinate system. Once established, the composite mapping enables direct transfer of gene-level and cell-type data from MERFISH into atlas space, allowing integration with other imaging and annotation datasets.

### DevCCF velocity flow transformation model
The Developmental Common Coordinate Framework (DevCCF)[16] provides a discrete set of age-specific templates that temporally sample the developmental trajectory. To model this biological progression more continuously, we introduce a velocity flow–based paradigm for inferring diffeomorphic transformations between developmental stages. This enables anatomically plausible estimation of intermediate templates or mappings at arbitrary timepoints between the E11.5 and P56 endpoints of the DevCCF. Our approach builds on established insights from time-varying diffeomorphic registration[65], where a velocity field governs the smooth deformation

 

of anatomical structures over time. Importantly, the framework is extensible and can naturally accommodate additional timepoints for the potential expansion of the DevCCF.

We first coalesced the anatomical labels across the seven DevCCF templates (E11.5, E13.5, E15.5, E18.5, P4, P14, P56) into 26 common structures that could be consistently identified across development. These include major brain regions such as the cortex, cerebellum, hippocampus, midbrain, and ventricles. For each successive pair of templates, we performed multi-label deformable registration using ANTsX to generate forward and inverse transforms between anatomical label volumes. From the P56 space, we randomly sampled approximately 1e6 points within and along the boundaries of each labeled region and propagated them through each pairwise mapping step (e.g., P56 → P14, P14 → P4, ..., E13.5 → E11.5). This procedure created time-indexed point sets tracing the spatial evolution of each region.

Using these point sets, we fit a continuous velocity field over developmental time using a generalized B-spline scattered data approximation method[86]. The field was parameterized over a log-scaled time axis to ensure finer temporal resolution during early embryonic stages, where morphological changes are most rapid. Optimization proceeded for approximately 125 iterations, minimizing the average Euclidean norm between transformed points at each step. Ten integration points were used to ensure numerical stability. The result is a smooth, differentiable vector field that defines a diffeomorphic transform between any two timepoints within the template range.

This velocity model can be used to estimate spatial transformations between any pair of developmental stages—even those for which no empirical template exists—allowing researchers to create interpolated atlases, align new datasets, or measure continuous structural changes. It also enables developmental alignment of multi-modal data (e.g., MRI to LSFM) by acting as a unifying spatiotemporal scaffold. The underlying components for velocity field fitting and integration are implemented in ITK, and the complete workflow is accessible in both ANTsPy (`ants.fit_time_varying_transform_to_point_sets(...)`) and ANTsR. In addition the availability of the DevCCF use case, self-contained examples and usage tutorials are provided in our public codebase.

### Automated brain extraction and parcellation with ANTsXNet

To support template-based deep learning approaches for structural brain extraction and parcellation, we implemented dedicated pipelines using the ANTsXNet framework. ANTsXNet comprises open-source deep learning libraries in both Python (ANTsPyNet) and R (ANTsRNet) that interface with the broader ANTsX ecosystem and are built on TensorFlow/Keras. Our mouse brain pipelines mirror existing ANTsXNet tools for human imaging but are adapted for species-specific anatomical variation, lower SNR, and heterogeneous acquisition protocols.

### Deep learning training setup

All network-based approaches were implemented using a standard U-net[87] architecture and hyperparameters previously evaluated in ANTsXNet pipelines for human brain imaging[45]. This design follows the 'no-new-net' principle[88], which demonstrates that a well-configured, conventional U-net can achieve robust and competitive performance across a wide range of biomedical segmentation tasks with little to no architectural modifications from the original. Both networks use a 3D U-net architecture implemented in TensorFlow/Keras, with five encoding/decoding levels and skip connections. The loss function combined Dice and categorical cross-entropy terms. Training used a batch size of 4, Adam optimizer with an initial learning rate of 2e-4, and early stopping based on validation loss. Training was performed on an NVIDIA DGX system (4 × Tesla V100 GPUs, 256 GB RAM). Model weights and preprocessing routines are shared across ANTsPyNet and ANTsRNet to ensure reproducibility and language portability. For both published and unpublished trained networks available through ANTsXNet, all training scripts and data augmentation generators are publicly available at https://github.com/ntustison/ANTsXNetTraining.

Robust data augmentation was critical to generalization across scanners, contrast types, and resolutions. We applied both intensity- and shape-based augmentation strategies:

- Intensity augmentations:

  - Gaussian, Poisson, and salt-and-pepper noise: `ants.add_noise_to_image(...)`
  - Simulated intensity inhomogeneity via bias field modeling[51]: `antspynet.simulate_bias_field(...)`
  - Histogram warping to simulate contrast variation[89]: `antspynet.histogram_warp_image_intensities(...)`

- Shape augmentations:

  - Random nonlinear deformations and affine transforms: `antspynet.randomly_transform_image_data(...)`
  - Anisotropic resampling across axial, sagittal, and coronal planes: `ants.resample_image(...)`

### Brain extraction

We originally trained a mouse-specific brain extraction model on two manually masked T2-weighted templates, generated from public datasets[67,68]. One of the templates was constructed from orthogonal 2D acquisitions using B-spline–based volumetric synthesis via `ants.fit_bspline_object_to_scattered_data(...)`. Normalized gradient magnitude was used as a weighting function to emphasize boundaries during reconstruction[86].

This training strategy provides strong spatial priors despite limited data by leveraging high-quality template images and aggressive augmentation to mimic population variability. During the development of this work, the network was further refined through community engagement. A user from a U.S.-based research institute applied this publicly available (but then unpublished) brain extraction tool to their own mouse MRI dataset. Based on feedback and iterative collaboration with the ANTsX team, the model was retrained and improved to better generalize to additional imaging contexts. This reflects our broader commitment to community-driven development and responsiveness to user needs across diverse mouse brain imaging scenarios.

The final trained network is available via ANTsXNet through the function `antspynet.mouse_brain_extraction(...)`. Additionally, both template/mask pairs are accessible via ANTsXNet. For example, one such image pair is available via:

- Template: `antspynet.get_antsxnet_data("bsplineT2MouseTemplate")`
- Brain mask: `antspynet.get_antsxnet_data("bsplineT2MouseTemplateBrainMask")`

### Brain parcellation

For brain parcellation, we trained a 3D U-net model using the DevCCF P56 T2-weighted template and anatomical segmentations derived from AllenCCFv3. This template-based training strategy enables the model to produce accurate, multi-region parcellations without requiring large-scale annotated subject data.

To normalize intensity across specimens, input images were preprocessed using rank-based intensity normalization (`ants.rank_intensity(...)`). Spatial harmonization was achieved through affine and deformable alignment of each extracted brain to the P56 template prior to inference. In addition to the normalized image input, the network also receives prior probability maps derived from the atlas segmentations, providing additional spatial context.

This general parcellation deep learning framework has also been applied in collaboration with other groups pursuing related but distinct projects. In one case, a model variant was adapted for T2-

weighted MRI using an alternative anatomical labeling scheme; in another, a separate model was developed for serial two-photon tomography (STPT) with a different parcellation set. All three models are accessible through a shared interface in ANTsXNet: `antspy-net.mouse_brain_parcellation(...)`. Ongoing work is further extending this approach to embryonic mouse brain data. These independent efforts reflect broader community interest in adaptable parcellation tools and reinforce the utility of ANTsXNet as a platform for reproducible, extensible deep learning workflows.

**Evaluation and reuse.** To assess model generalizability, both the brain extraction and parcellation networks were evaluated on an independent longitudinal dataset comprising multiple imaging sessions with varied acquisition parameters[69]. Although each label or imaging modality required retraining, the process was streamlined by the reusable ANTsX infrastructure enabled by rapid adaptation with minimal overhead. These results illustrate the practical benefits of a template-based, low-shot strategy and modular deep learning framework. All trained models, associated training scripts, and supporting resources are openly available and designed for straightforward integration into ANTsX workflows.

### Reporting summary
Further information on research design is available in the Nature Portfolio Reporting Summary linked to this article.

## Data availability
The following datasets were used in this study and are publicly available: • Allen Common Coordinate Framework (AllenCCFv3): Available from the Allen Institute for Brain Science at https://atlas.brain-map.org/atlas. • Developmental Common Coordinate Framework (DevCCF) MRI and LSFM datasets: Publicly available via the Kim Lab https://kimlab.io/home/projects/DevCCF/index.html. • MERFISH spatial transcriptomics data: Previously published[46] https://portal.brain-map.org. • Developmental datasets for brain extraction and segmentation: – High-resolution MRI data of brain C57BL/6 and BTBR mice in three different anatomical views: https://data.mendeley.com/datasets/dz9x23fttt/1. – CAMRI Mouse Brain Data: https://openneuro.org/datasets/ds002868/versions/1.0.1• Evaluation dataset for brain extraction and segmentation: A longitudinal microstructural MRI dataset in healthy C57Bl/6 mice at 9.4 Tesla https://www.frdr-dfdr.ca/repo/dataset/9ea832ad-7f36-4e37-b7ac-47167c0001c1. • ANTsXNet-pretrained templates and models: Available through ANTsPy at https://github.com/ANTsX/ANTsPyNet. Source data are provided with this paper.

## Code availability
All processing pipelines and supporting code are openly available at:
•https://github.com/ntustison/ANTsXMouseBrainMapping(DevCCF velocity model and deep learning parcellation). Also contains the text, scripts, and data to reproduce the manuscript (including figures).
•https://github.com/dontminchenit/CCFAlignmentToolkit(MERFISH and fMOST workflows).

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

## Acknowledgements

Support for the research reported in this work includes funding from the National Institute of Biomedical Imaging and Bioengineering (RO1-EB031722) and National Institute of Mental Health (RF1-MH124605, U24-MH114827, and NIH RF1MH124605 to Y.K.). We also acknowledge the data contribution of Dr. Adam Raikes (GitHub @araikes) of the Center for Innovation in Brain Science at the University of Arizona for refining the weights of the mouse brain extraction network.

## Author contributions

N.T., M.C., and J.G. wrote the main manuscript text and figures. M.C., M.K., R.D., S.S., Q.W., L.N., J.D., C.G., and J.G. developed the Allen registration pipelines. N.T., F.K., J.G., and Y.K. developed the time-varying velocity transformation model for the DevCCF. N.T. and M.T. developed the brain parcellation and cortical thickness methodology. All authors reviewed the manuscript.

## Competing interests

The authors declare no competing interests.
