## [Transparent Peer Review file · Nature Communications]

The ANTsX ecosystem for mapping the mouse brain

Corresponding Author: Professor James Gee

Version 0:

Reviewer comments:

Reviewer #1

(Remarks to the Author)

Summary:

The paper "Modular strategies for spatial mapping of diverse cell type data of the mouse brain" presents four ANTsX-based workflows to map diverse mouse-brain data into common coordinate frameworks. These include: (1) aligning MERFISH spatial transcriptomic data to the Allen CCFv3, (2) registering high-resolution fMOST structural volumes, and (3) mapping developing brain MRI and LSFM images into a continuous developmental atlas (DevCCF) via a novel velocity flow model. Additionally, the authors introduce deep learning models for mouse brain extraction and parcellation trained with minimal annotated data ("single-shot" and "two-shot" learning).

The paper's central contribution lies in demonstrating how flexible combinations of existing ANTsX tools can be applied robustly across diverse data modalities. Two of the presented pipelines have been published previously, while the remaining two are new contributions. Notably, the introduction of the DevCCF velocity flow model fills a critical gap by enabling continuous, biologically informed interpolation across developmental stages. The low-shot CNN models for brain extraction and parcellation demonstrate that high-quality mappings can be achieved even with minimal annotated data, reducing the barrier for broader use. The primary novelty is in the technological packaging and dissemination of mapping pipelines that are openly accessible and reproducible by the community - critical for ongoing efforts in data harmonisation and multimodal integration. Overall, this paper establishes a solid foundation for further integration of molecular, structural, and developmental data and provides an important contribution to the community.

While the presented pipelines are practical and relevant, several aspects require clarification or further development. The paper attempts to serve both as a resource of general guidance for data mapping and as a vehicle for presenting new pipelines, which dilutes focus. It lacks a structured section offering general mapping recommendations across modalities. Despite claiming modularity and robustness, the pipelines appear to require significant dataset-specific tuning. Moreover, external validation is limited, and there are no benchmarks against existing tools. The evaluation metrics, particularly for MERFISH and DevCCF, are sparse or subjective. Addressing these concerns would significantly strengthen the paper's clarity and impact.

Major comments

1. Clarify the dual focus of the paper.

As noted in the summary, the paper appears to pursue two goals: (1) to provide general guidance for spatial registration of neuroimaging data, and (2) to introduce two new mapping pipelines. The abstract and introduction imply the former is the main focus, but this is not clearly developed in the body of the manuscript. I strongly recommend the authors clearly state in the introduction which pipelines are new, which are previously published (and where). For instance, in Section 2.1.2, it is unclear whether the approach and dataset have been previously published.

2. Add a general guidance section

To support the aim of helping researchers choose appropriate mapping strategies across modalities, I suggest including a "general guidance" section (possibly as the first results section). This could include a matrix or decision tree linking data modality to pipeline type, including discussion of when to use affine, deformable, label-based, or DL-based mapping. Pros and cons of these approaches should be compared and discussed. This would make the paper significantly more useful as a reference for non-expert users.

3. Re-evaluate the claims of modularity and robustness

While the workflows reuse common ANTsX components, many parts are still heavily tailored to specific modalities. There is minimal discussion of how robust these pipelines are to hyperparameters (e.g. CNN training settings, registration smoothing, label selection) or to imaging artefacts such as tears and distortions. Furthermore, each evaluation is based on in-house datasets; no external test cases are used to demonstrate generalizability. I suggest the authors provide guidance on parameter selection and include robustness evaluations across a wider range of imaging conditions or sample types.

4. Strengthen the evaluations and include benchmarks

The evaluations currently lack detail and comparisons to existing tools. I recommend:

4.2. MERFISH mapping: Include a benchmark against STalign, which performs image-based registration of MERFISH data to atlas space.

4.2 DevCCF velocity flow model: Compare to traditional pairwise interpolation between timepoints. Include biological validation—e.g., do interpolated brains at non-template ages yield plausible anatomy?

4.3 Mouse brain mapping: Evaluate CNN robustness across different scanners or MR contrasts. In Fig. 9c, the DL parcellation method appears advantaged compared to image-based registration because it uses label-driven optimization, and Dice scores may be inflated. Use manually segmented data as a reference for a fairer comparison.

5. Clarify and revise terminology

* The terms “single-shot” and “two-shot” are misleading. In machine learning, they refer to test-time adaptation with minimal data, not the number of annotated templates used in training. “Low-template training” or “template-augmented training” would be clearer.

* The term “velocity field” is not introduced or defined in the main text. A brief, intuitive explanation should be added early in the relevant section.

Minor Comments

1. Title

The title emphasizes “cell type data,” but only the MERFISH dataset contains transcriptomic cell-type information. This is misleading. A more accurate title might refer to “molecular and morphological data” or simply “multi-modal mouse brain data.”

2. Section 1.1

Clarify that the focus is on image-based mapping strategies. For MERFISH and other ST methods, alternative mapping approaches exist (like Tangram) and should be acknowledged.

3. Line 234

Clarify how evaluation is performed. What does it mean that “only seven small subregions were missed”?

4. Line 277

What does “similar quantitative assessment” refer to? Please indicate where the results of that assessment are shown.

5. Line 305

Add a brief explanation of “velocity field” (e.g., “a smoothly varying deformation model that describes how anatomical structures change over developmental time”).

6. Line 330

Clarify what is meant by “we used 28 days for the P56 data.”

7. Figure 9

This figure is not referenced in the text. Clarify whether Fig. 9c shows one subject or an aggregate over multiple brains.

8. Line 660

How were cells assigned to regions? Since regions consist of multiple cell types, this could impact interpretation.

9. Line 727

Please include details on network architecture and training parameters in the methods section itself. Relying solely on internet links is not ideal for long-term reproducibility.

(Remarks on code availability)

The GitHub repository contains code and data for the two novel contributions, i.e. the velocity flow model for the DevCCF and the CNN-based brain extraction and mapping methods. The README file is nicely structured and contains all relevant code for the DevCCF, but no code snippets are contained for the CNN-based brain mapping methods. Although there are some training python scripts in the repository, it is not clear to me how to use them to reproduce the brain mapping. There are links to AntsPyNet functions that implement the mouse brain mapping, but it would be clearer to include the actual code lines in the README as well.

From the README I believe it would be possible (with some effort) to reproduce the two novel contributions, although I did not attempt to install and run the code. In the GitHub, there is no mention of the other pipelines presented in the paper.

Reviewer #2

(Remarks to the Author)

This paper presents a high detail view of three recent alignment applications, their use in recent publications and how they were implemented in the ANTsX framework.

The material is well presented and is likely of strong interest to researchers doing brain work that requires similar mouse brain alignment (especially developmental, and especially those hoping to do this alignment in ANTsX) but impact seems somewhat confined to those who have read the methods sections of the related papers and are still hungry for more. I only

re-reviewed the relevant methods sections for the Developmental Mouse CCF to compare them to this MS and there is significant value added by the longer form discussion, and greater computational detail here, however the content in the current MS never quite comes together to address the larger issues of modularity and multi modal alignment brought up in the abstract.

The modularity of the approaches seems to be a claim based largely on their being built on top of ANTsX. While building on top of a well developed library is an aid to modularity, it certainly doesn't guarantee it in either the software engineering or algorithmic sense. More general relevance might be aided by more explicitly drawing out how (other than by building on ANTsX) the described work provides insight on either the algorithmic modularity of alignment or the engineering of modular software libraries. (e.g. an ontology of building blocks to aid their uniform interoperability) These are of course important questions of wide interest, but they seem to not be specifically addressed here.

(Remarks on code availability)

I quickly skimmed the code in <https://github.com/dontminchenit/CCFAlignmentToolkit> and <https://github.com/dontminchenit/CCFAlignmentToolkit> without trying to run it or understand the structure in great detail.

Though this is about par for scientific software I note the absence of high level code overview and limited info in the readme.md. None of the .py files I probed at random appeared to have any comments documenting what they were for. None of this is unusual, but given the context a narrative code overview in the documentation would be a nice complement to the MS and a big aid to anyone trying to apply the methods.

FYI. the readme.md and setup.txt files in CCFAlignmentToolkit look incomplete.

Reviewer #3

(Remarks to the Author)

In this manuscript, Tustison et al. propose several improvements for the open source image registration framework ANTsX. They cover several data types and workflow related to atlas registration for brain imaging. They include workflow for registering separately published spatial transcriptomic (MERFISH) and high-resolution morphology (fMOST) data onto the Allen Common Coordinate Framework (AllenCCFv3), and developmental (MRI and LSFM) data into the Developmental Common Coordinate Framework (DevCCF). They specifically detail two new technical contributions to the ANTsX framework: an interpolation of developmental trajectories in-between the timepoints available in the DevCCF using the integration of estimated flow fields, and a brain MRI parcelisation network.

- On the MERFISH workflow: this allows to describe in more detail the processing workflow of data otherwise published and is interesting and useful.

- On fMOST workflow: it is not entirely clear whether and how the described workflow has been used and published elsewhere or if this is a wholly original contribution.

On both cases there is a significant amount of details in methods that would help potential users, on top of the code itself. The evaluations are extremely short and amounts to little more than 'it works'.

- The DevCCF velocity flow transformation model address a significant need in the use of developmental atlases and the proposed solution is, as far as we can tell, elegant and working. it is pretty frustrating however as a lot of detail and discussion are missing that would have been expected on a full paper on the topic. Presumably this is not the first attempted to solve such a problem, yet a bibliography on the subject is very sparse. Technical and mathematical details are sparse as well, beyond a few paragraph of text, and there is no evaluation to speak of.

To be used in practice one would need not only the technical assurance of convergence of an algorithm, but also some discussion of the biological relevance of the interpolation. The development of a brain is not done by warping space... In particular, one would expect the ontology of the parcelation to change significantly across development beyond the 13 common areas picked for correspondance, and I am not sure I saw a discussion on converting/interpolating parcelations. Can we actually use the resulting method to make measures on more than those 13 regions for examples?

- The last part on automatic parcelation again address a very important topic in practice with a clean and well documented solution.

Not being an expert in MRI I found this part less clear than the others, with maybe some vocabulary that is common in that community being used without context. To pick one 'brain extraction' is I guess segmentation of the brain area from the image that may contain other part (skull, etc) but this is not defined. Is the brain extraction network then performing a semantic segmentation, ie outputting a mask in image space of where the brain is?

Same for parcelation itself: what exactly is the 'brain parcelation' task in this context? Multilabel semantic segmentation with a very reduced set of preselected brain regions? Again the section reads more as a technical documentation than an article describing a new method. Evaluation on alternative datasets is great but the description is a bit sparse. Figure 9 is not cited in the text and the paragraph that would refer to it, the last one before the discussion, is very sparse. It seem to use the learnt parcelation network to cue another registration step, with the final region delimitation being evaluated, but this is not very

explicit, and anyway a lot of things to happen in the very last paragraph, supposedly on evaluation, of a paper.

Overall this manuscript relates several advanced use cases of, and significant addition to, the ANTsX brain registration framework. The problem tackled are of major importance in the community and the solutions proposed are very interesting and useful. The fact that every result shown can be redone from code to data is excellent and somehow legitimates the paper as a showcase of ANTsX progress and use.

On the other end a lot of details one would expect of scientific paper on numerical data analysis methods are missing. Whether or not that is a major issue can be discussed. Scientific open source software need academic publications as this is the de facto currency in academic circles, and they are often left to the methods of biological papers, with even less details than here, so this format can be interesting in that context.

Eventually I do think that to warrant publication in Nature Communications, a generic journal, to advertise the ANTsX framework, some work toward being more accessible by a wider audience would be beneficial.

typo:
'brian' l390

(Remarks on code availability)

I did not review the code per se for lack of time but could check that it is as complete as advertised. Looking at it answered some questions I had reading the paper...

Reviewer #4

(Remarks to the Author)

This manuscript presents a framework using the ANTsX toolkit to map a range of mouse brain datasets — including MERFISH spatial transcriptomics, fMOST high-resolution morphology, and developmental MRI/LSFM — into common coordinate frameworks. The authors describe both shared and modality-specific solutions, introduce a velocity flow model for developmental interpolation, and propose a pipeline for brain parcellation. Overall, the manuscript is well-written and likely to be of value to the mouse brain imaging community. However, I have some concerns and questions. Addressing these concerns could strengthen the work.

Major Comments:

- **Validation of Velocity Model:** The validation of the velocity flow model is primarily convergence-based, without quantitative assessment of predictive performance. The authors should perform some quantitative assessment, for example a leave-one-timepoint-out cross-validation to test whether the model can accurately interpolate held-out stages (e.g., predicting P04 from E11.5–E18.5 and P14–P56). Evaluation using anatomical landmarks or region overlaps would significantly strengthen this component.
- **Biological Plausibility of Developmental Transforms (Figure 5):** Some of the transformations depicted in Figure 5 — especially those involving extrapolation from late to early stages — appear anatomically implausible (e.g., the P56 transformations result in unusual structural features). Please discuss the biological validity of such transformations and consider constraining interpolation to neighbouring timepoints in line with developmental continuity.

(Remarks on code availability)

All described code and data appear to be available.

Version 1:

Reviewer comments:

Reviewer #1

(Remarks to the Author)

The revised manuscript addresses all my comments. The focus of the manuscript and its novel contributions are now clearly delineated. The additional discussion on evaluation choices is appreciated, and terminology is now very clear. With this revision, the manuscript will be a valuable resource for researchers using the ANTsX ecosystem to map mouse brain data.

Below my response to the authors response:

Major comments:

1. The revised abstract and introduction greatly clarify the focus of the manuscript.
2. I understand that adding a general guidance section is out of scope with the current focus of the manuscript on detailing for mapping strategies using ANTsX.
- 3.+4. **Evaluations:** Thank you for the clarification. The segmentation method is indeed evaluated on an external dataset. However, the other pipelines (as far as I understand) are mostly qualitatively evaluated. I appreciate the extended comments in the discussion to this effect and think that the evaluation suffices in the context of this paper. However, I believe that investing efforts into building benchmarking datasets and advanced metrics will greatly benefit the community, as solely

relying on qualitative expert evaluations of pipelines makes comparisons across pipelines very difficult.
Benchmark comparisons: I appreciate the effort the authors took to make a comparison with existing tools for the MERFISH mapping pipeline.

5. My concerns about terminology are addressed in the revised manuscript.

Minor comments: All my minor comments are addressed.

I do agree with the authors that providing code details in github repositories is crucial for reproducibility, however omitting any model details in the paper makes it hard for readers to understand the approaches from the paper alone. I appreciate the added details about model and training parameters in the manuscript for this purpose.

(Remarks on code availability)

The README's of both repositories are now much clearer and will greatly aid researches aiming to use the pipelines presented in the manuscript.

Reviewer #2

(Remarks to the Author)

The authors revisions have made it significantly easier to understand the relationship between the alignment methods that are a more detailed presentation of technical components from prior work and novel computational methods (temporal warping and segmentation). The methods appear sound and of general interest. Though no individual component is of major novelty collectively, they cover a nice range of case studies in things you might want to do.

The documentation of the source code appears much improved, and I think would be sufficient to reproduce results without excessive forensic examination of the source code.

I do feel that revisions haven't addressed my concerns about the centrality of modularity claims to the title and abstract, when in fact modularity is not addressed. Minimally to support this claim at least the four applications covered in this work should be discussed in terms of shared modules (within ANTsX, and ideally in terms of the scripts themselves which might/should share levels of abstraction above ANTsX to support this claim). The emphasis should be on the shared (algorithmic and software) components of these tasks. The MS seems to have the opposite emphasis, on the diversity of ANTsX functionality utilized to achieve these quite diverse pipelines. This is a worthy goal but it feels like the real emphasis of the MS could be better reflected in an alternate title if more substantial handling of modularity is out of scope.

(Remarks on code availability)

I re reviewed the documentation for all 4 pipelines and spot checked a few of the scripts, though I did not try to build out an environment and actually run them. Revised documentation seems like it will be helpful in guiding users through the scripts

Reviewer #3

(Remarks to the Author)

The authors did a thorough job of reworking their manuscript and adding the necessary content to address my comments and I have no other comments.

I believe this is a useful and well written contribution that will help the community to use complex registration pipeline.

(Remarks on code availability)

I have not reviewed the code again.

Reviewer #4

(Remarks to the Author)

The authors have addressed my concerns/comments.

(Remarks on code availability)

Nature Communications manuscript NCOMMS-25-23244-T

We sincerely thank the reviewers and editors for their thoughtful and constructive feedback on our manuscript. We are encouraged by the Reviewers' recognition of the manuscript's contributions. We also appreciate the reviewers' detailed suggestions to improve clarity, methodological depth, and accessibility. In response, we have significantly revised the manuscript to clarify its dual aims, strengthen evaluation of the proposed novel methods, and improve both biological interpretability and user guidance. This revision also includes significantly reducing the number of words to conform with manuscript guidelines. We have also expanded code documentation and updated terminology to ensure broader accessibility across disciplines. Below, we provide a point-by-point response and detail the corresponding changes made to the manuscript.

Reviewer #1

Remarks to the Author

Summary:

The paper "Modular strategies for spatial mapping of diverse cell type data of the mouse brain" presents four ANTsX-based workflows to map diverse mouse-brain data into common coordinate frameworks. These include: (1) aligning MERFISH spatial transcriptomic data to the Allen CCFv3, (2) registering high-resolution fMOST structural volumes, and (3) mapping developing brain MRI and LSFM images into a continuous developmental atlas (DevCCF) via a novel velocity flow model. Additionally, the authors introduce deep learning models for mouse brain extraction and parcellation trained with minimal annotated data ("single-shot" and "two-shot" learning).

The papers central contribution lies in demonstrating how flexible combinations of existing ANTsX tools can applied robustly across diverse data modalities. Two of the presented pipelines have been published previously, while the remaining two are new contributions. Notably, the introduction of the DevCCF velocity flow model fills a critical gap by enabling continuous, biologically informed interpolation across developmental stages. The low-shot CNN models for brain extraction and parcellation demonstrate that high-quality mappings can be achieved even with minimal annotated data, reducing the barrier for broader use. The primary novelty is in the technological packaging and dissemination of mapping pipelines that are openly accessible and reproducible by the community - critical for ongoing efforts in data harmonisation and multimodal integration. Overall, this paper establishes a solid foundation for further integration of molecular, structural, and developmental data and provides an important contribution to the community.

While the presented pipelines are practical and relevant, several aspects require clarification or further development. The paper attempts to serve both as a resource of general guidance for

data mapping and as a vehicle for presenting new pipelines, which dilutes focus. It lacks a structured section offering general mapping recommendations across modalities. Despite claiming modularity and robustness, the pipelines appear to require significant dataset-specific tuning. Moreover, external validation is limited, and there are no benchmarks against existing tools. The evaluation metrics, particularly for MERFISH and DevCCF, are sparse or subjective. Addressing these concerns would significantly strengthen the paper's clarity and impact.

We thank the reviewer for the careful assessment of our manuscript. We are especially grateful for the recognition of our contributions and for the suggestions to improve clarity, evaluation rigor, and overall focus. In response, we have revised the manuscript to explicitly distinguish new contributions from previously published workflows, added guidance for selecting appropriate mapping strategies across modalities, and strengthened evaluation and validation metrics for the MERFISH alignment and the DevCCF velocity model.

Major comments

1. Clarify the dual focus of the paper.

As noted in the summary, the paper appears to pursue two goals: (1) to provide general guidance for spatial registration of neuroimaging data, and (2) to introduce two new mapping pipelines. The abstract and introduction imply the former is the main focus, but this is not clearly developed in the body of the manuscript. I strongly recommend the authors clearly state in the introduction which pipelines are new, which are previously published (and where). For instance, in Section 2.1.2, it is unclear whether the approach and dataset have been previously published.

We appreciate the reviewer's careful reading and agree that the dual focus of the manuscript warranted clearer framing. We have revised the Abstract and Introduction to address this directly and to distinguish which components are novel. In addition, we emphasize that the MERFISH and fMOST workflows, while presented in reproducible form here, are adapted from earlier collaborative ANTsX-based efforts rather than previously as explicitly published pipelines. The revised Introduction now reads:

Building on this third category, we describe a set of modular, ANTsX-based pipelines specifically tailored for mapping diverse mouse brain data into standardized anatomical frameworks. These include two new pipelines: a velocity field-based interpolation model that enables continuous transformations across developmental timepoints of the DevCCF, and a template-based deep learning pipeline for whole brain segmentation (i.e., brain extraction) and structural anatomical regional labeling of the brain (i.e., brain parcellation) requiring minimal annotated data. In addition, we include two modular pipelines for aligning MERFISH and fMOST datasets to the Allen CCFv3. While the MERFISH dataset was previously published as part of earlier BICCN efforts⁴⁶, the full image processing and registration workflow had not been described in detail until now. The fMOST workflow, by contrast, was developed internally to support high-resolution morphology mapping and has not been previously published in any form. Both pipelines were built using ANTsX tools, adapted for collaborative use with the Allen Institute, and are now released as fully reproducible, open-source workflows to support reuse and extension by the community. To facilitate broader adoption, we also provide general guidance for customizing these strategies across imaging modalities and data types. We first introduce key components

of the ANTsX toolkit, which provide a basis for all of the mapping workflows described here, and then detail the specific contributions made in each pipeline. (lines 82-99)

2. Add a general guidance section

To support the aim of helping researchers choose appropriate mapping strategies across modalities, I suggest including a “general guidance” section (possibly as the first results section). This could include a matrix or decision tree linking data modality to pipeline type, including discussion of when to use affine, deformable, label-based, or DL-based mapping. Pros and cons of these approaches should be compared and discussed. This would make the paper significantly more useful as a reference for non-expert users.

We appreciate the reviewer’s insightful suggestion and agree with the motivation that accessible guidance on mapping strategy selection would be valuable for many readers. However, developing a comprehensive decision tree or modality-to-method matrix of the type requested would require a detailed treatment of the theoretical and practical trade-offs across several mapping methods. In our view, such a contribution would be better suited to a focused tutorial or review article on image registration rather than a methods-focused research paper introducing specific pipelines. That said, we have made several changes in the revised manuscript to better support readers who may be seeking high-level guidance. For instance, the revised Introduction explicitly states:

These tools span multiple classes of mapping problems: cross-modality image registration, landmark-driven alignment, temporal interpolation across developmental stages, and deep learning-based segmentation. As such, they also serve as illustrative case studies for adapting ANTsX tools to other use cases. (lines 117–120)

To further assist readers, we provide a high-level overview of relevant ANTsX functionality in the Methods section, organized by task:

Although focused on distinct data types, the three pipelines presented in this work share common components that address general challenges in mapping mouse cell type data. These include intensity correction, image denoising, registration, template generation, and visualization. Table 1 provides a concise summary of the relevant ANTsX functionality. (lines 425–429)

Each pipeline section (MERFISH, fMOST, velocity flows for the DevCCF, and brain extraction and parcellation) now includes clear explanations of why specific strategies were chosen (e.g., when label-based registration was favored over intensity-based registration due to modality differences, or how intermediate atlases help mitigate domain shifts). These revisions offer practical, modality-specific insight into method selection while remaining consistent with the goals of the manuscript.

In addition, for users seeking interactive support, we highlight the publicly available ANTsX tutorial site, which walks through core functionality and application design patterns. The various ANTsX ecosystem GitHub forums also serve as an active venue for community exchange, and have directly contributed to the evolution of several components featured in

this work, including improvements to the brain extraction and parcellation tools discussed in this manuscript. For example, detailing the development of the brain extraction workflow, we write:

This training strategy provides strong spatial priors despite limited data by leveraging high-quality template images and aggressive augmentation to mimic population variability. During the development of this work, the network was further refined through community engagement. A user from a U.S.-based research institute applied this publicly available (but then unpublished) brain extraction tool to their own mouse MRI dataset. Based on feedback and iterative collaboration with the ANTsX team, the model was retrained and improved to better generalize to additional imaging contexts. This reflects our broader commitment to community-driven development and responsiveness to user needs across diverse mouse brain imaging scenarios. (lines 611-619)

We hope these resources, together with the expanded manuscript guidance, help balance technical specificity with accessibility for both expert and non-expert readers.

3. Re-evaluate the claims of modularity and robustness

While the workflows reuse common ANTsX components, many parts are still heavily tailored to specific modalities. There is minimal discussion of how robust these pipelines are to hyperparameters (e.g. CNN training settings, registration smoothing, label selection) or to imaging artefacts such as tears and distortions. Furthermore, each evaluation is based on in-house datasets; no external test cases are used to demonstrate generalizability. I suggest the authors provide guidance on parameter selection and include robustness evaluations across a wider range of imaging conditions or sample types.

We appreciate the reviewer's attention to robustness and generalizability. We address each concern below:

1. Evaluation on independent data:

We respectfully clarify that the segmentation pipelines were evaluated using an independent, publicly available dataset not used in training or even having been seen prior to its discovery specifically for the purpose of evaluation. This dataset originates from a publicly released study on age-related brain changes in mice and was not used during model development. In addition to age variation, it includes realistic heterogeneity in scanning protocols, providing a meaningful test of generalization. This is emphasized in the revised manuscript:

For evaluation, we used an additional publicly available dataset⁷⁰ that is completely independent from the data used in training the brain extraction and parcellation networks. (lines 341-342)

and in the Methods section we now write:

To assess model generalizability, both brain extraction and parcellation models were evaluated on an independent longitudinal dataset with varied acquisition parameters⁷⁰. (lines 647-649)

2. Robustness to hyperparameters (CNN training settings, label selection, etc.):

The CNN models were built using standardized U-net architectures and training parameters, consistent with best practices from the ANTsXNet platform. As we now state in the revised Methods section:

All network-based approaches were implemented using a standard U-net⁸⁸ architecture and hyperparameters previously evaluated in ANTsXNet pipelines for human brain imaging⁴⁵. This design follows the 'no-new-net' principle⁸⁹, which demonstrates that a well-configured, conventional U-net can achieve robust and competitive performance across a wide range of biomedical segmentation tasks with little to no architectural modifications from the original. (lines 576-580)

This principle supports our strategy of combining aggressive data augmentation with standardized architectures to handle domain shifts and limited training data. While we acknowledge that we did not systematically vary hyperparameters, our goal is to promote broadly usable tools based on reproducible defaults already validated across domains. We highlight that the proposed approach was successfully adapted by external ANTsX users to novel datasets and integrated into available ANTsX tools. For instance, community members have built and applied these segmentation tools to mouse-specific imaging applications:

- *Whole-brain segmentation,*
- *T2-weighted MRI with alternative label sets,*
- *Serial two-photon tomography (STPT) data with alternative label sets, and*
- *Ongoing efforts in embryonic mouse brain segmentation.*

We have detailed examples of such user interactions within the manuscript to demonstrate that these experiences suggest a high degree of robustness to domain shifts without extensive tuning.

3. Claims of modularity vs. modality-specific tailoring:

We agree that individual pipelines contain modality-specific design elements. However, for the reported work, this tailoring occurs within a modularized framework (i.e., ANTsX) that enables reuse of core tools such as preprocessing routines, template generation, registration components, and deep learning utilities across data types. This modularity is emphasized in the Introduction:

Finally, general-purpose toolkits such as elastix, Slicer3D, and the Advanced Normalization Tools Ecosystem (ANTsX) have all been applied to mouse brain mapping scenarios. These toolkits support modular workflows that can be flexibly composed from reusable components, offering a powerful alternative to rigid, modality-specific solutions. However, their use often requires familiarity with pipeline modules, parameter tuning, and tool-specific conventions which can limit adoption.

Building on this third category, we describe a set of modular, ANTsX-based pipelines specifically tailored for mapping diverse mouse brain data into standardized anatomical frameworks (lines 76-84).

We view this process of tailoring via modular reuse as a strength, not a limitation. It allows researchers to quickly assemble or adapt workflows for their specific needs while maintaining consistency and reusability.

4. Imaging artifacts (e.g., distortions, tears):

Although we do not explicitly model all possible artifacts, we do address several real-world imaging issues in the pipelines. Examples include:

- *Stripe artifact removal for fMOST data via frequency-domain filtering,*
- *Intensity inhomogeneity correction via N4 bias field correction,*
- *Label smoothing and interpolation for misalignment resilience in sectioned MERFISH data, and*
- *Learned label interpolation in anisotropic MRI data.*

Where possible, the use of anatomical labels also helps reduce sensitivity to intensity-based noise or artifacts and provides potential separate avenues for alignment (e.g., see the ANTsPy tool `label_image_registration(...)`).

4. Strengthen the evaluations and include benchmarks

The evaluations currently lack detail and comparisons to existing tools. I recommend:

4.2. MERFISH mapping: Include a benchmark against STalign, which performs image-based registration of MERFISH data to atlas space.

We thank the reviewer for the suggestion to benchmark our MERFISH registration workflow against STalign. We recognize STalign as an important contribution to the field and appreciate its methodological innovation in aligning spatial transcriptomic data to a specified atlas space. We explored this recommendation by attempting to apply STalign to our MERFISH dataset; however, we encountered several technical challenges in reproducing the workflow with our specific data. These technical issues were ultimately documented in a recent (August 20, 2025) GitHub issue (Issue #55) which is still pending a response.

While we were ultimately unable to complete a direct benchmark, we continue to view STalign as a valuable approach and would welcome future comparisons should compatibility improve. In the meantime, we provide a fully reproducible, open pipeline based on the ANTsX framework, which may offer complementary benefits in terms of modularity and integration with other imaging pipelines.

4.2 DevCCF velocity flow model: Compare to traditional pairwise interpolation between timepoints. Include biological validation—e.g., do interpolated brains at non-template ages yield plausible anatomy?

We appreciate this thoughtful suggestion. Compared to traditional pairwise interpolation, our velocity flow model computes a single, smooth, time-continuous deformation field which enables direct mapping between any two points along the developmental axis. We now clarify this distinction in the Results section:

Unlike traditional pairwise interpolation, which requires sequential warping through each intermediate stage, this model, defined by a time-varying velocity field (i.e., a smooth vector field defined over space and time that governs the continuous deformation of an image domain), allows direct computation of deformations between any two time points in the continuum which improves smoothness and enables flexible spatiotemporal alignment. (lines 224-229)

Regarding biological plausibility, informal expert evaluation (co-author Yongsoo Kim) of long-range transformations (e.g., P56 to E11.5 sampled continuously) revealed smooth but occasionally diverging from known developmental patterns (e.g., hippocampus). Rather than indicating model failure, this highlighted how the learned trajectory reflects the input data and can help guide future atlas development. We now include this in the Discussion:

Interestingly, long-range transformations (e.g., P56 to E11.5) revealed anatomy evolving in plausible ways yet sometimes diverging from known developmental patterns (e.g., hippocampal shape changes) reflecting the input data and offering insight into temporal gaps. These behaviors could assist future efforts to determine which additional time points would most improve spatiotemporal coverage.” (lines 398-402)

Finally, we point the Reviewer to the relevant GitHub repository with improved documentation and annotation where this work can be completely reproduced as it includes the relevant code, data, and commentary to guide the interested user.

4.3 Mouse brain mapping: Evaluate CNN robustness across different scanners or MR contrasts. In Fig. 9c, the DL parcellation method appears advantaged compared to image-based registration because it uses label-driven optimization, and Dice scores may be inflated. Use manually segmented data as a reference for a fairer comparison.

We thank the reviewer for this important suggestion. In the Evaluation, we used the predicted labels from the deep learning model solely to drive the registration to the AllenCCFv3 atlas space. However, the Dice scores in Figure 9(c) were computed using the manually segmented labels, which were independently transformed into the same atlas space for comparison. This approach ensures that the evaluation reflects anatomical correspondence rather than any circularity from the model’s own output. We have clarified this point in the figure caption and the main text to avoid confusion and to emphasize the fairness of the comparison between deep learning–guided and intensity-only registration strategies.

5. Clarify and revise terminology

* The terms “single-shot” and “two-shot” are misleading. In machine learning, they refer to test-time adaptation with minimal data, not the number of annotated templates used in training. “Low-template training” or “template-augmented training” would be clearer.

We thank the reviewer for pointing out the difficulty with the usage of the terms “single-shot” and “two-shot.” To address this concern, we have removed all use of “single-shot” and “two-shot” from the manuscript and GitHub documentation. In their place, we use clearer terminology such as “template-based training” and describe the number and nature of the annotated examples explicitly where relevant. For example:

We originally trained a mouse-specific brain extraction model on two manually masked T2-weighted templates, generated from public datasets.^{68,69} (lines 606-607)

These changes are reflected throughout the revised manuscript and in the corresponding GitHub repository, which now includes working examples for brain extraction, brain parcellation, and DevCCF-based mapping.

* The term “velocity field” is not introduced or defined in the main text. A brief, intuitive explanation should be added early in the relevant section.

We thank the reviewer for this helpful suggestion. In the revised manuscript, we now include an intuitive explanation of the velocity field concept early in the DevCCF section to aid reader understanding. Specifically, we state:

Unlike traditional pairwise interpolation, which requires sequential warping through each intermediate stage, this model, defined by a time-varying velocity field (i.e., a smooth vector field defined over space and time that governs the continuous deformation of an image domain), allows direct computation of deformations between any two time points in the continuum which improves smoothness and enables flexible spatiotemporal alignment. (lines 224-229)

Minor Comments

1. Title

The title emphasizes “cell type data,” but only the MERFISH dataset contains transcriptomic cell-type information. This is misleading. A more accurate title might refer to “molecular and morphological data” or simply “multi-modal mouse brain data.”

*We appreciate the suggestion and have revised the title to “**Modular strategies for spatial mapping of multi-modal mouse brain data**” to better reflect the molecular and morphological scope.*

2. Section 1.1

Clarify that the focus is on image-based mapping strategies. For MERFISH and other ST methods, alternative mapping approaches exist (like Tangram) and should be acknowledged.

We thank the reviewer for this important suggestion. We have clarified in the revised manuscript that our focus is on image-based spatial mapping strategies, particularly within the context of anatomical alignment and template registration. As noted in the updated text:

Also, while alternative strategies for mapping single-cell spatial transcriptomic data exist (e.g., gene expression–based models such as Tangram²⁴) this work focuses on image-based anatomical alignment to common coordinate frameworks using spatially resolved reference images. (lines 61-64)

3. Line 234

Clarify how evaluation is performed. What does it mean that “only seven small subregions were missed”?

We thank the reviewer for pointing out this lack of clarity. Evaluation was performed by manually analyzing the MERFISH cell locations with respect to the Allen CCF after registration. The 7 missing regions refer to the small regions in the Allen CCF annotations where no cells were found post registration. This analysis was performed as part of a previous publication (Yao et al 2023). We have modified the text to better express this:

As previously reported⁴⁶, further assessment of the alignment showed that, of the 554 terminal regions (gray matter only in the AllenCCFv3), only seven small subregions did not contain cells from the MERFISH dataset post registration: frontal pole, layer 1 (FRP1), FRP2/3, FRP5; accessory olfactory bulb, glomerular layer (AOBgl); accessory olfactory bulb, granular layer (AOBgr); accessory olfactory bulb, mitral layer (AOBmi); and accessory supraoptic group (ASO). A broader discussion of evaluation design choices and evaluation rationale is included in the Discussion. (lines 171-178)

More generally, we have expanded the evaluation approach for both the fMOST and MERFISH data in the Discussion:

As part of collaborative efforts with the Allen Institute for Brain Science and the broader BICCN initiative, we developed two modular pipelines for mapping MERFISH and fMOST datasets to the AllenCCFv3. These workflows were designed to accommodate the specific requirements of high-resolution transcriptomic and morphological data while leveraging reusable components from the ANTsX ecosystem. The MERFISH pipeline incorporates preprocessing and registration steps tailored to known anatomical and imaging artifacts in multiplexed spatial transcriptomic data. While the general mapping strategy is applicable to other sectioned histological datasets, these refinements demonstrate how general-purpose tools can be customized to meet the demands of specialized modalities. The fMOST workflow, in contrast, emphasizes reusability and consistency across large datasets. It introduces an intermediate, canonical fMOST atlas to stabilize transformations to the AllenCCFv3, reducing the need for repeated manual alignment and enabling standardized mapping of single-neuron reconstructions to a common coordinate framework.

Evaluation of both workflows followed established QA/QC protocols used at the Allen Institute, emphasizing biologically meaningful criteria such as expected gene-marker alignment (MERFISH) and accurate reconstruction of neuronal morphology (fMOST). These domain-informed assessments, also used in prior large-scale mapping projects⁴⁶, prioritize

task-relevant accuracy over other possible benchmarks such as Dice coefficients or landmark distances. While formal quantitative scores are not reported here, both pipelines have demonstrated reliable, expert-validated performance in collaborative contexts. Additional documentation and evaluation commentary are available in the updated CCFAlignmentToolkit GitHub repository. (lines 368-389)

4. Line 277

What does “similar quantitative assessment” refer to? Please indicate where the results of that assessment are shown.

*We agree with the reviewer that this sentence lacked clarity, but respectively note that line 277 in the manuscript indicated “similar **qualitative** assessment” (emphasis added). This referred to qualitative analysis our anatomists performed to ensure that single-cell neuron reconstructions from the fMOST data were adequately aligned after registration. Unfortunately, it is currently not feasible to perform quantitative analysis on the alignment of these reconstructions because the Allen CCF does not contain analogous single-cell neuron reconstructions to serve as ground truth. It is also not clear yet how one might establish ground truth correspondence for single neuron alignment between brains. Since this analysis was qualitative and refers to fMOST single-cell neuron reconstruction methods and data outside the scope of this paper, we have opted to remove this sentence to reduce confusion.*

5. Line 305

Add a brief explanation of “velocity field” (e.g., “a smoothly varying deformation model that describes how anatomical structures change over developmental time”).

As suggested previously by the Reviewer, we added an intuitive explanation of the “velocity field” in the subsection previous to Line 305, “a smooth vector field defined over space and time that governs the continuous deformation of an image domain.” Immediately in the subsequent paragraph, to provide the reader with a concrete idea of the instantiation of the velocity field we added:

The velocity field is represented as a 4D ITK image where each voxel stores the x,y,z components of motion at a given time point. Integration of the time-varying velocity field uses 4th order Runge-Kutta (ants.integrate_velocity_field(...))⁸¹. (lines 231–234)

6. Line 330

Clarify what is meant by “we used 28 days for the P56 data.”

We clarified this ambiguity as follows:

Within this logarithmic temporal transform, P56 was assigned a span of 28 postnatal days to reflect known developmental dynamics (i.e., in terms of modeling the continuous deformation, the morphological changes between Day 28 and Day 56 are insignificant). This improved the temporal distribution of integration points (Figure 4, right panel). (lines 262-266)

7. Figure 9

This figure is not referenced in the text. Clarify whether Fig. 9c shows one subject or an aggregate over multiple brains.

We thank the reviewer for catching this omission. In the revised manuscript, we have added a reference to Figure 9 in the Results section of the segmentation pipelines. Specifically, we now clarify:

Figure 8 summarizes the whole-brain overlap between manually segmented reference masks and the predicted segmentations for all 84 images in the evaluation cohort. The proposed network demonstrates excellent performance in brain extraction across a wide age range. To further assess the utility of the parcellation network, we used the predicted labels to guide anatomically informed registration to the AllenCCFv3 atlas using ANTsX multi-component registration, and compared this to intensity-only registration (Figure 9). While intensity-based alignment performs reasonably well, incorporating the predicted parcellation significantly improves regional correspondence. Dice scores shown in Figure 9(c) were computed using manually segmented labels transformed to AllenCCFv3 space. (lines 347-355)

We further clarify in the caption of Figure 9:

(c) Dice overlap scores across the full evaluation cohort (n=84), comparing anatomical alignment achieved via registration using intensity alone versus registration guided by the predicted parcellation. Dice values were computed using manually segmented labels transformed to AllenCCFv3 space.

8. Line 660

How were cells assigned to regions? Since regions consist of multiple cell types, this could impact interpretation.

We thank the reviewer for suggesting this clarification. Regions in the MERFISH data were assigned using a cell type clustering approach previously detailed in (Yao et al 2023). We have expanded the relevant text to provide a better summary of this approach:

Label creation. To assign region labels to the MERFISH data, we use a cell type clustering approach previously detailed (Yao et al 2023). In short, manually dissected scRNAseq data was used to establish the distribution of cell types present in each of the following major regions: cerebellum, CTXsp, hindbrain, HPF, hypothalamus, isocortex, LSX, midbrain, OLF, PAL, sAMY, STRd, STRv, thalamus and hindbrain. Clusters in the scRNA-seq dataset were then used to assign similar clusters of cell types in the MERFISH data to the regions they are predominantly found in the scRNA-seq data. To account for clusters that were found at low frequency in regions outside its main region we calculated for each cell its 50 nearest neighbors in physical space and reassigned each cell to the region annotation dominating its neighborhood. (lines 498-507)

9. Line 727

Please include details on network architecture and training parameters in the methods section itself. Relying solely on internet links is not ideal for long-term reproducibility.

We thank the Reviewer for this recommendation. To improve long-term reproducibility and transparency, we have updated the Methods section to include a brief summary of the network architecture and training parameters used in the brain extraction and structural labeling pipelines:

Both networks use a 3D U-net architecture implemented in TensorFlow/Keras, with five encoding/decoding levels and skip connections. The loss function combined Dice and categorical cross-entropy terms. Training used a batch size of 4, Adam optimizer with an initial learning rate of $2e-4$, and early stopping based on validation loss. (lines 581-584)

While we agree that essential details should be included in the manuscript text, we respectfully note that open-source repositories such as GitHub are widely used in scientific research precisely to support long-term reproducibility, transparency, and maintenance. In fact, persistent, version-controlled repositories often offer a more durable and verifiable record of software and model provenance than manuscript text alone, which often cannot capture full implementation details.

Remarks on code availability

The GitHub repository contains code and data for the two novel contributions, i.e. the velocity flow model for the DevCCF and the CNN-based brain extraction and mapping methods. The README file is nicely structured and contains all relevant code for the DevCCF, but no code snippets are contained for the CNN-based brain mapping methods. Although there are some training python scripts in the repository, it is not clear to me how to use them to reproduce the brain mapping. There are links to AntsPyNet functions that implement the mouse brain mapping, but it would be clearer to include the actual code lines in the README as well.

From the README I believe it would be possible (with some effort) to reproduce the two novel contributions, although I did not attempt to install and run the code. In the GitHub, there is no mention of the other pipelines presented in the paper.

We appreciate the reviewer's careful assessment of the code availability and agree that clarity and accessibility are essential for community uptake. In response, we have significantly revised the GitHub README to directly address the concerns raised. Specifically, we have:

- *Added self-contained code examples demonstrating how to use the ANTsXNet-based pipelines for brain extraction and regional labeling. These include both `ANTsPyNet` and `ANTsRNet` usage, with copyable command-line code and clear documentation of expected inputs and outputs.*
- *Removed ambiguous terminology such as "single-shot" and "two-shot" in favor of more accurate descriptions following the reviewer's earlier suggestions.*

- *Clarified the scope of this repository which is focused on the novel contributions of this paper, specifically the DevCCF velocity flow model and the structural segmentation pipelines. As such, the repository includes all code and data needed to reproduce these results, including a documented pipeline for velocity field optimization and pretrained ANTsXNet models for segmentation.*

We note that the fMOST and MERFISH mapping pipelines presented in the manuscript are developed and maintained separately in the CCF Alignment Toolkit GitHub repository which is publicly available and linked in the manuscript. These pipelines were contributed by the second author as part of earlier BICCN work. In response to the reviewers' feedback, this repository has been updated with improved documentation and usage guidance.

- *The repository now includes explicit narrative descriptions for both the one-time atlas construction workflow (e.g., building an fMOST template and aligning it to Allen CCFv3) and the routine runtime scripts. This includes clear explanations of key scripts like `antsMultivariateTemplateConstruction2.sh`, `AtlasToCCFSequentialRegistration.py`, `RegisterfMOSTtoCCFGlobal.py`, and how they are used in the pipeline .*
- *Script inputs and outputs are now well-specified and documented—for example, what files are needed (fMOSTTemplate, label files) and what outputs are produced (e.g., forward/inverse warp fields, registered images), making it more user-friendly for new adopters.*

Reviewer #2

Remarks to the Author

This paper presents a high detail view of three recent alignment applications, their use in recent publications and how they were implemented in the ANTsX framework.

The material is well presented and is likely of strong interest to researchers doing brain work that requires similar mouse brain alignment (especially developmental, and especially those hoping to do this alignment in ANTsX) but impact seems somewhat confined to those who have read the methods sections of the related papers and are still hungry for more. I only re-reviewed the relevant methods sections for the Developmental Mouse CCF to compare them to this MS and there is significant value added by the longer form discussion, and greater computational detail here, however the content in the current MS never quite comes together to address the larger issues of modularity and multi modal alignment brought up in the abstract.

The modularity of the approaches seems to be a claim based largely on their being built on top of ANTsX. While building on top of a well developed library is an aid to modularity, it certainly doesn't guarantee it in either the software engineering or algorithmic sense. More general relevance might be aided by more explicitly drawing out how (other than by building on ANTsX) the described work provides insight on either the algorithmic modularity of alignment or the

engineering of modular software libraries. (e.g. an ontology of building blocks to aid their uniform interoperability) These are of course important questions of wide interest, but they seem to not be specifically addressed here.

We thank the reviewer for their thoughtful evaluation and recognition of the value added by this manuscript, especially in extending and contextualizing methods from related publications. We are particularly encouraged by the reviewer’s observations that the paper provides a “high detail view” with “greater computational detail” and is likely of “strong interest to researchers doing brain work that requires similar mouse brain alignment.” Regarding clearer articulation of our claims regarding modularity and multimodal alignment, we agree that simply building on a flexible framework such as ANTsX does not, in itself, ensure modularity in the engineering or algorithmic sense. Our intent was not to present a formal treatment of software modularity, but rather to highlight how reusable components in ANTsX can be flexibly assembled and adapted to address diverse mapping challenges across modalities through concrete, fully reproducible pipelines. That said, we appreciate the reviewer’s call to clarify and strengthen our claims. In response, we have revised the manuscript to better articulate the basis for modularity in both the algorithmic and engineering senses:

- *We have revised the Introduction to clarify that “modularity” in our context refers to components within a flexible, cross-language software ecosystem:*

To facilitate broader adoption, we also provide general guidance for customizing these strategies across imaging modalities and data types. We first introduce key components of the ANTsX toolkit, which provide a basis for all of the mapping workflows described here, and then detail the specific contributions made in each pipeline. (lines 95-99)

and

Building on this third category, we describe a set of modular, ANTsX-based pipelines specifically tailored for mapping diverse mouse brain data into standardized anatomical frameworks. (lines 82-84)

- *We now emphasize in the Methods section how each pipeline draws from a shared ontology of processing stages (e.g., preprocessing, registration), and how this conceptual structure facilitates adaptation to new use cases:*

Although focused on distinct data types, the three pipelines presented in this work share common components that address general challenges in mapping mouse brain data. These include correcting image intensity artifacts, denoising, spatial registration, template generation, and visualization. Table 1 provides a concise summary of the relevant ANTsX functionality. (lines 425-429)

- *We revised the Discussion to more explicitly connect our design choices to the idea of composable workflows. For example, we now state:*

The ANTsX ecosystem offers a powerful foundation for constructing scalable, reproducible pipelines for mouse brain data analysis. Its modular design and multi-platform support enable researchers to develop customized workflows without extensive new software development. (lines 411-414)

Finally, while we agree that a full ontology of building blocks could be highly beneficial, we believe that would be better served in a focused review or software design manuscript. Nevertheless, we hope that the detailed examples and accompanying code repositories provide a clear and reusable blueprint for investigators seeking to compose their own modular workflows in ANTsX.

Remarks on code availability

I quickly skimmed the code in <https://github.com/dontminchenit/CCFAlignmentToolkit> and <https://github.com/dontminchenit/CCFAlignmentToolkit> without trying to run it or understand the structure in great detail.

Though this is about par for scientific software I note the absence of high level code overview and limited info in the readme.md. None of the .py files I probed at random appeared to have any comments documenting what they were for. None of this is unusual, but given the context a narrative code overview in the documentation would be a nice complement to the MS and a big aid to anyone trying to apply the methods.

FYI. the readme.md and setup.txt files in CCFAlignmentToolkit look incomplete.

We thank the reviewer for their thoughtful assessment of the available code and documentation. As noted in our response to Reviewer 1, the CCFAlignmentToolkit repository (<https://github.com/dontminchenit/CCFAlignmentToolkit>) was contributed by the second author as part of earlier BICCN work and supports the MERFISH and fMOST pipelines. In response to the reviewer's feedback, this repository has been updated with improved documentation and usage guidance.

We also wish to highlight that the two novel contributions described in this manuscript (i.e., the DevCCF velocity flow model and the template-based deep learning parcellation tools) are supported by a separate repository (<https://github.com/ntustison/ANTsXMouseBrainMapping>) developed and maintained by the first author.

In light of the reviewer's comment on modularity and clarity, we also encourage readers in the manuscript to consult the broader ANTsX tutorial (<https://tinyurl.com/antsxtutorial>), which provides structured examples of ANTsX component usage and is likewise maintained by the first author.

Reviewer #3

Remarks to the Author

In this manuscript, Tustison et al. propose several improvements for the open source image registration framework ANTsX. They cover several data types and workflow related to atlas registration for brain imaging. They include workflow for registering separately published spatial transcriptomic (MERFISH) and high-resolution morphology (fMOST) data onto the Allen Common Coordinate Framework (AllenCCFv3), and developmental (MRI and LSFM) data into the Developmental Common Coordinate Framework (DevCCF). They specifically detail two new technical contributions to the ANTsX framework: an interpolation of developmental trajectories in-between the timepoints available in the DevCCF using the integration of estimated flow fields, and a brain MRI parcelisation network.

We thank the reviewer for the helpful summary and for recognizing the range of workflows and technical contributions presented in the manuscript.

- On the MERFISH workflow: this allows to describe in more detail the processing workflow of data otherwise published and is interesting and useful.

- On fMOST workflow: it is not entirely clear whether and how the described workflow has been used and published elsewhere or if this is a wholly original contribution.

We appreciate the reviewer's comments regarding the MERFISH and fMOST workflows. In response, we have clarified the origin and publication status of each pipeline, as well as their relationship to prior BICCN work.

In addition, we include two modular pipelines for aligning MERFISH and fMOST datasets to the Allen CCFv3. While the MERFISH dataset was previously published as part of earlier BICCN efforts⁴⁶, the full image processing and registration workflow had not been described in detail until now. The fMOST workflow, by contrast, was developed internally to support high-resolution morphology mapping and has not been previously published in any form. Both pipelines were built using ANTsX tools, adapted for collaborative use with the Allen Institute, and are now released as fully reproducible, open-source workflows to support reuse and extension by the community. (lines 88-95)

On both cases there is a significant amount of details in methods that would help potential users, on top of the code itself. The evaluations are extremely short and amounts to little more than 'it works'.

We appreciate the reviewer's recognition of the level of methodological detail provided in the manuscript and in the accompanying code repositories. We also thank the reviewer for candidly highlighting the limited scope of the evaluations as originally presented. In retrospect, we agree that our initial explanation may have been too brief, and we understand how it could have come across as "it just works." We take responsibility for that lack of clarity and have revised the manuscript to better contextualize our evaluation strategy.

As noted in the revised text, both the MERFISH and fMOST pipelines were developed in close collaboration with the Allen Institute for Brain Science as part of broader BICCN efforts. The workflows were validated internally through established QA/QC protocols used at the Allen Institute, protocols that have been applied in previous large-scale mapping projects (e.g., Yao et al., Nature 2023, cited in the manuscript). These include domain-expert evaluation of anatomical alignment, such as consistency of gene marker expression patterns (MERFISH) or successful mapping of single-cell neuronal morphologies (fMOST). While such assessments may not align with conventional quantitative benchmarks like landmark distances or Dice coefficients, they reflect practical standards used in collaborative neuroimaging projects and are aligned with how image registration quality is often judged in applied research settings.

That said, we agree that a more explicit explanation is warranted. Towards this end we added at the end of the Evaluation subsections for both fMOST and MERFISH the following:

A broader discussion of evaluation design choices and validation rationale is included in the Discussion. (lines 176-178, 211-212)

In the Discussion, the relevant two paragraphs read as follows:

As part of collaborative efforts with the Allen Institute for Brain Science and the broader BICCN initiative, we developed two modular pipelines for mapping MERFISH and fMOST datasets to the AllenCCFv3. These workflows were designed to accommodate the specific requirements of high-resolution transcriptomic and morphological data while leveraging reusable components from the ANTsX ecosystem. The MERFISH pipeline incorporates preprocessing and registration steps tailored to known anatomical and imaging artifacts in multiplexed spatial transcriptomic data. While the general mapping strategy is applicable to other sectioned histological datasets, these refinements demonstrate how general-purpose tools can be customized to meet the demands of specialized modalities. The fMOST workflow, in contrast, emphasizes reusability and consistency across large datasets. It introduces an intermediate, canonical fMOST atlas to stabilize transformations to the AllenCCFv3, reducing the need for repeated manual alignment and enabling standardized mapping of single-neuron reconstructions to a common coordinate framework.

Evaluation of both workflows followed established QA/QC protocols used at the Allen Institute, emphasizing biologically meaningful criteria such as expected gene-marker alignment (MERFISH) and accurate reconstruction of neuronal morphology (fMOST). These domain-informed assessments, also used in prior large-scale mapping projects⁴⁶, prioritize task-relevant accuracy over other possible benchmarks such as Dice coefficients or landmark distances. While formal quantitative scores are not reported here, both pipelines have demonstrated reliable, expert-validated performance in collaborative contexts. Additional documentation and evaluation commentary are available in the updated CCFAlignmentToolkit GitHub repository. (lines 368-389)

We hope that the clarified rationale in the manuscript, along with the release of reproducible workflows, strikes a useful balance between transparency, utility, and the practical realities of evaluation in cross-institutional collaborative settings.

- The DevCCF velocity flow transformation model address a significant need in the use of developmental atlases and the proposed solution is, as far as we can tell, elegant and working. it is pretty frustrating however as a lot of detail and discussion are missing that would have been expected on a full paper on the topic. Presumably this is not the first attempted to solve such a problem, yet a bibliography on the subject is very sparse. Technical and mathematical details are sparse as well, beyond a few paragraph of text, and there is no evaluation to speak of.

To be used in practice one would need not only the technical assurance of convergence of an algorithm, but also some discussion of the biological relevance of the interpolation. The development of a brain is not done by warping space... In particular, one would expect the ontology of the parcelation to change significantly across development beyond the 13 common areas picked for correspondance, and I am not sure I saw a discussion on converting/interpolating parcelations. Can we actually use the resulting method to make measures on more than those 13 regions for examples?

We appreciate the reviewer's concern that brain development is not a simple spatial warping process and agree that any interpolated anatomical trajectory should be interpreted cautiously. Our goal was not to prescribe a biologically "correct" transformation, but rather to provide a flexible, data-driven model of continuous deformation that enables interpolation and visualization across developmental stages. In this sense, the velocity model serves as an exploratory tool that can highlight both the strengths and gaps in the underlying developmental data. As discussed in our response to Reviewer 1, the model differs from traditional pairwise interpolation by computing a single, smooth, and longitudinally continuous deformation field that allows direct mapping between any two points along the developmental axis.

Concerning the reviewer's specific questions regarding the 13 common regions, we have also expanded the Discussion to describe how, relatedly, an informal expert evaluation (co-author Yongsoo Kim) revealed that long-range interpolations (e.g., from P56 to E11.5) produce reasonably plausible transformations, but sometimes diverge from known developmental trajectories, particularly in complex regions such as the hippocampus.

Interestingly, long-range transformations (e.g., P56 to E11.5) revealed anatomy evolving in plausible ways yet sometimes diverging from known developmental patterns (e.g., hippocampal shape changes) reflecting the input data and offering insight into temporal gaps. These behaviors could assist future efforts to determine which additional time points would most improve spatiotemporal coverage (lines 398-402).

Rather than indicating model failure, these observations provide insight into, for example, where additional templates would improve the time points comprising the DevCCF. We believe this type of model-based interrogation (e.g., using the framework to ask where the developmental trajectory is under-constrained) is a valuable feature and one of the motivations for our proposed pipeline.

- The last part on automatic parcelation again address a very important topic in practice with a clean and well documented solution.

Not being an expert in MRI I found this part less clear than the others, with maybe some vocabulary that is common in that community being used without context. To pick one 'brain extraction' is I guess segmentation of the brain area from the image that may contain other parts (skull, etc) but this is not defined. Is the brain extraction network then performing a semantic segmentation, i.e. outputting a mask in image space of where the brain is?

Same for parcellation itself: what exactly is the 'brain parcellation' task in this context? Multilabel semantic segmentation with a very reduced set of preselected brain regions? Again the section reads more as a technical documentation than an article describing a new method. Evaluation on alternative datasets is great but the description is a bit sparse. Figure 9 is not cited in the text and the paragraph that would refer to it, the last one before the discussion, is very sparse. It seems to use the learnt parcellation network to cue another registration step, with the final region delineation being evaluated, but this is not very explicit, and anyway a lot of things to happen in the very last paragraph, supposedly on evaluation, of a paper.

We thank the reviewer for their detailed comments on the brain extraction and parcellation components and the assessment that this section presents a "clean and well-documented solution" to an important problem in practice. We also appreciate the suggestions to improve clarity for readers less familiar with MRI terminology, and have revised the manuscript accordingly. Specifically, in the Introduction we have explicitly defined "extraction" and "parcellation":

...and a template-based deep learning pipeline for whole brain segmentation (i.e., brain extraction) and structural anatomical regional labeling of the brain (i.e., brain parcellation) requiring minimal annotated data. (lines 85-88)

To improve clarity of the brain parcellation section, we have revised this paragraph to explicitly cite Figure 9 in the main text and clarify that the parcellation outputs were used to inform a second-stage label-guided registration step:

Figure 8 summarizes the whole-brain overlap between manually segmented reference masks and the predicted segmentations for all 84 images in the evaluation cohort. The proposed network demonstrates excellent performance in brain extraction across a wide age range. To further assess the utility of the parcellation network, we used the predicted labels to guide anatomically informed registration to the AllenCCFv3 atlas using ANTsX multi-component registration, and compared this to intensity-only registration (Figure 9). While intensity-based alignment performs reasonably well, incorporating the predicted parcellation significantly improves regional correspondence. Dice scores shown in Figure 9(c) were computed using manually segmented labels transformed to AllenCCFv3 space (lines 347-355).

Overall this manuscript relates several advanced use cases of, and significant addition to, the ANTsX brain registration framework. The problems tackled are of major importance in the community and the solutions proposed are very interesting and useful. The fact that every result shown can be redone from code to data is excellent and somehow legitimates the paper as a showcase of ANTsX progress and use.

On the other end a lot of details one would expect of scientific paper on numerical data analysis methods are missing. Whether or not that is a major issue can be discussed. Scientific open source software need academic publications as this is the de facto currency in academic circle, and they are often left to the methods of biological papers, with even less details than here, so this format can be interesting in that context.

Eventually I do think that to warrant publication in Nature Communication, a generic journal, to advertise the ANTsX framework, some work toward being more accessible by a wider audience would be beneficial.

We thank the reviewer for their thoughtful and generous assessment. One of our primary goals was to offer the community a set of modular, executable workflows built on ANTsX that address real-world mapping challenges across modalities and developmental time. We also appreciate the reviewer's observation that the manuscript sits at the intersection of scientific software documentation and methods reporting, and we agree that greater accessibility is important. As detailed in the general introduction to our response, we have revised the manuscript throughout to improve clarity for a wider audience, including more intuitive definitions of domain-specific terms (e.g., brain extraction, parcellation, velocity field), clearer articulation of workflow objectives and evaluation strategies, and emphasis on the practical utility of each pipeline, and how individual components can be reused or adapted. We hope these revisions help make the manuscript both a useful technical reference and a more accessible entry point for users new to ANTsX or to spatial mapping of mouse brain data.

typo:

'brian' l390

Fixed.

Remarks on code availability

I did not review the code per se for lack of time but could check that is it as complete as advertised. Looking at it answered some questions I had reading the paper...

We appreciate that the reviewer found it useful. Please see the revised and extended commentary in the advertised repositories in response to the comments made by the reviewers.

Reviewer #4

Remarks to the Author

This manuscript presents a framework using the ANTsX toolkit to map a range of mouse brain datasets — including MERFISH spatial transcriptomics, fMOST high-resolution morphology, and developmental MRI/LSFM — into common coordinate frameworks. The authors describe both

shared and modality-specific solutions, introduce a velocity flow model for developmental interpolation, and propose a pipeline for brain parcellation. Overall, the manuscript is well-written and likely to be of value to the mouse brain imaging community. However, I have some concerns and questions. Addressing these concerns could strengthen the work.

We thank the reviewer for their thoughtful assessment and for recognizing the value of the manuscript to the mouse brain imaging community.

Major Comments:

- Validation of Velocity Model: The validation of the velocity flow model is primarily convergence-based, without quantitative assessment of predictive performance. The authors should perform some quantitative assessment, for example a leave-one-timepoint-out cross-validation to test whether the model can accurately interpolate held-out stages (e.g., predicting P04 from E11.5–E18.5 and P14–P56). Evaluation using anatomical landmarks or region overlaps would significantly strengthen this component.

We thank the reviewer for this thoughtful suggestion. In response, we have added a validation comparing the transformations modeled by the velocity flow approach to the transforms derived by conventional pairwise registration (SyN from ANTsX). Specifically, for each internal time point, we warp regional labels from adjacent timepoints using both approaches, apply atlas-based labeling, and compute Dice overlap scores. The results show that the velocity model achieves comparable segmentation accuracy to SyN-based interpolation, supporting its utility as a smooth and flexible alternative. These results are now included in Section 2 and Figure 4.

- Biological Plausibility of Developmental Transforms (Figure 5): Some of the transformations depicted in Figure 5 — especially those involving extrapolation from late to early stages — appear anatomically implausible (e.g., the P56 transformations result in unusual structural features). Please discuss the biological validity of such transformations and consider constraining interpolation to neighbouring timepoints in line with developmental continuity.

We appreciate the reviewer's observations regarding the biological plausibility of long-range interpolations, particularly extrapolations from late to early stages such as P56 to E11.5. As noted in the revised Discussion and in response to earlier reviewers, we agree that some of these transformations deviate from expected developmental trajectories, especially in complex regions such as the hippocampus. These observations were confirmed by one of the co-authors (Yongsoo Kim, senior author of the DevCCF), who identified plausible yet biologically inconsistent transitions during visual inspection of interpolated volumes. As we write in the revised Discussion:

Interestingly, long-range transformations (e.g., P56 to E11.5) revealed anatomy evolving in plausible ways yet sometimes diverging from known developmental patterns (e.g., hippocampal shape changes) reflecting the input data and offering insight into temporal gaps. These behaviors could assist future efforts to determine which additional time points would most improve spatiotemporal coverage (lines 398-402).

Rather than viewing these deviations as a failure of the velocity flow model, we interpret them as a valuable outcome of the framework. Because the model provides a smooth, queryable trajectory across time, it enables researchers to visualize and interrogate under-constrained or anatomically uncertain regions of the DevCCF. In this way, the model serves not only as an interpolation tool but also as a guide for identifying where additional templates or refined anatomical priors are needed. While downstream users may choose to constrain interpolation to neighboring stages in practice, we believe the current framework supports flexible and exploratory analysis of developmental dynamics in a principled way.

Remarks on code availability

All described code and data appear to be available.

Please see the revised and extended commentary in the advertised repositories in response to the comments made by the reviewers.

Nature Communications manuscript NCOMMS-25-23244-T

We thank the reviewers and editors for their thoughtful evaluations and positive assessment of our revised manuscript. Below we address the remaining minor points raised in this round of reviews.

Reviewer #1

Remarks to the Author:

The revised manuscript addresses all my comments. The focus of the manuscript and its novel contributions are now clearly delineated. The additional discussion on evaluation choices is appreciated, and terminology is now very clear. With this revision, the manuscript will be a valuable resource for researches using the ANTsX ecosystem to map mouse brain data.

Below my response to the authors response:

Major comments:

1. The revised abstract and introduction greatly clarify the focus of the manuscript.
2. I understand that adding a general guidance section is out of scope with the current focus of the manuscript on detailing for mapping strategies using ANTsX.
- 3.+4. Evaluations: Thank you for the clarification. The segmentation method is indeed evaluated on an external dataset. However, the other pipelines (as far as I understand) are mostly qualitatively evaluated. I appreciate the extended comments in the discussion to this effect and think that the evaluation suffices in the context of this paper. However, I believe that investing efforts into building benchmarking datasets and advanced metrics will greatly benefit the community, as solely relying on qualitative expert evaluations of pipelines makes comparisons across pipelines very difficult.

Benchmark comparisons: I appreciate the effort the authors took to make a comparison with existing tools for the MERFISH mapping pipeline.

5. My concerns about terminology are addressed in the revised manuscript.

Minor comments: All my minor comments are addressed.

I do agree with the authors that providing code details in github repositories is crucial for reproducibility, however omitting any model details in the paper makes it hard for readers to understand the approaches from the paper alone. I appreciate the added details about model and training parameters in the manuscript for this purpose.

Remarks on code availability:

The README's of both repositories are now much clearer and will greatly aid researches aiming to use the pipelines presented in the manuscript.

We appreciate the reviewer's positive feedback.

Reviewer #2

Remarks to the Author:

The authors revisions have made it significantly easier to understand the relationship between the alignment methods that are a more detailed presentation of technical components from prior work and novel computational methods (temporal warping and segmentation). The methods appear sound and of general interest. Though no individual component is of major novelty collectively, they cover a nice range of case studies in things you might want to do.

The documentation of the source code appears much improved, and I think would be sufficient to reproduce results without excessive forensic examination of the source code.

I do feel that revisions haven't addressed my concerns about the centrality of modularity claims to the title and abstract, when in fact modularity is not addressed. Minimally to support this claim at least the four applications covered in this work should be discussed in terms of shared modules (within ANTsX, and ideally in terms of the scripts themselves which might/should share levels of abstraction above ANTsX to support this claim). The emphasis should be on the shared (algorithmic and software) components of these tasks. The MS seems to have the opposite emphasis, on the diversity of ANTsX functionality utilized to achieve these quite diverse pipelines. This is a worthy goal but it feels like the real emphasis of the MS could be better reflected in an alternate title if more substantial handling of modularity is out of scope.

We revised the title to "The ANTsX ecosystem for mapping the mouse brain" to better reflect the manuscript's focus on reproducible ANTsX-based workflows rather than conceptual modularity. Corresponding minor edits were made to ensure consistency.

Remarks on code availability:

I re reviewed the documentation for all 4 pipelines and spot checked a few of the scripts, though I did not try to build out an environment and actually run them. Revised documentation seems like it will be helpful in guiding users through the scripts

We appreciate the reviewer's positive feedback.

Reviewer #3

Remarks to the Author:

The authors did a thorough job of reworking their manuscript and adding the necessary content to address my comments and I have no other comments.

I believe this is a useful and well written contribution that will help the community to use complex registration pipeline.

We appreciate the reviewer's positive feedback.

Remarks on code availability:

I have not reviewed the code again.

Reviewer #4

Remarks to the Author:

The authors have addressed my concerns/comments.

We appreciate the reviewer for confirming that no further revisions were needed.